# RISE: Radius of Influence based Subgraph Extraction for 3D Molecular Graph Explanation

**Jingxiang Qu** [* 1]  **Wenhan Gao** [* 2]  **Jiaxing Zhang** [3]  **Xufeng Liu** [1]  **Hua Wei** [4]  **Haibin Ling** [1]  **Yi Liu** [2 1]

## Abstract

3D Geometric Graph Neural Networks (GNNs) have emerged as transformative tools for modeling molecular data. Despite their predictive power, these models often suffer from limited interpretability, raising concerns for scientific applications that require reliable and transparent insights. While existing methods have primarily focused on explaining molecular substructures in 2D GNNs, the transition to 3D GNNs introduces unique challenges, such as handling the implicit dense edge structures created by a cut-off radius. To tackle this, we introduce a novel explanation method specifically designed for 3D GNNs, which localizes the explanation to the immediate neighborhood of each node within the 3D space. Each node is assigned a *radius of influence*, defining the localized region within which message passing captures spatial and structural interactions crucial for the model's predictions. This method leverages the spatial and geometric characteristics inherent in 3D graphs. By constraining the subgraph to a localized radius of influence, the approach not only enhances interpretability but also aligns with the physical and structural dependencies typical of 3D graph applications, such as molecular learning. The code is available at https://github.com/QuJX/RISE.

## 1. Introduction

With recent advances in deep learning, molecular learning has become a pivotal field of research, driving significant progress in drug discovery, protein engineering, and materials science (Zhang et al., 2025; Bronstein et al., 2021). Traditionally, molecules have commonly been modeled as 2D planar graphs (Gao et al., 2021; Liu et al., 2021; 2020), with atoms represented as nodes and chemical bonds as edges, disregarding their geometric configurations. However, it is widely believed that chemical behaviors and biological functions of molecules are largely determined by their 3D geometric structures (Tusnády & Simon, 1998; Hansch & Fujita, 1964). The limitations of 2D representations in capturing geometric information have led to a growing focus on 3D graph learning (Gainza et al., 2020; Townshend et al., 2021). Consequently, 3D graph neural networks (GNNs) with geometric features have demonstrated superior performance and thus become dominant across various tasks (Schütt et al., 2017; Gasteiger et al., 2020; Liu et al., 2022; Wang et al., 2022b; Brandstetter et al., 2022; Subedi et al., 2024; Yan et al., 2022; Wang et al., 2023).

Although 3D GNNs have gained appealing performance in molecular graph learning, they are largely regarded as black-box models. GNN explanation methods seek to find crucial information from a molecular graph that significantly impacts the model's predictions. The main goal is to identify a compact subset of edges (a subgraph) that accurately captures the behavior of the original graph (Ying et al., 2019; Yuan et al., 2020; Luo et al., 2020; Yuan et al., 2021; Huang et al., 2023; Liu et al., 2025). Chemists can verify whether these explanatory subgraphs align with actual chemical substructures, such as functional groups and motifs, that contribute the most to the chemical properties. While these explanation methods have demonstrated effectiveness for 2D GNNs, they overlook the fundamental distinctions of 3D GNNs, leading to explanations with inferior fidelity.

There are two major differences between 2D and 3D GNNs that should be taken into account when designing explanation methods for 3D molecular learning. **(1)** *Difference in Representation*: In 3D molecular graphs, the edges are no longer actual chemical bonds but are instead constructed based on cut-off distances (Schütt et al., 2017; Gasteiger et al., 2020; Liu et al., 2022), resulting in a rapid increase of edges; **(2)** *Difference in Learning*: 3D GNNs aim to model both bonded and non-bonded interactions (e.g., elec-

---

[*]Equal contribution [1]Department of Computer Science, Stony Brook University, Stony Brook, NY, United States [2]Department of Applied Mathematics & Statistics, Stony Brook University, Stony Brook, NY, United States [3]Department of Informatics, New Jersey Institute of Technology, Newark, NJ, United States [4]School of Computing and Augmented Intelligence, Arizona State University, Tempe, AZ, United States. Correspondence to: Yi Liu <yi.liu.4@stonybrook.edu>.

*Proceedings of the 42$^{nd}$ International Conference on Machine Learning*, Vancouver, Canada. PMLR 267, 2025. Copyright 2025 by the author(s).

trostatics, hydrogen bonds) through distances and spatial modeling. ***These differences lead to a broader search space and significant difficulty in optimization for existing explanation methods, which will be discussed further in Sec. 3.1.1. More importantly, these characteristics render the explanatory results from existing methods chemically uninterpretable.*** In 2D molecular graph learning, edges directly represent chemical bonds, which naturally define explanatory substructures such as functional groups (Wu et al., 2018; Diao et al., 2022; Ma et al., 2020). However, this direct correspondence no longer holds in 3D GNNs and the optimized substructures often consist of a set of "random edges" that are chemically uninterpretable. As a result, there is a pressing need for a principled explanation framework that considers the differences and closes this gap.

In order to design principled 3D GNN explanation methods considering these differences, a natural question arises:

> ***(Q)*** *In atomic systems, interactions between atoms or nodes often weaken with distance. Does the 3D GNN inherently adhere to this physical principle in its learning process? If so, can we design explanation methods by referring to distances?*

In particular, we find that the proximity (distances between atoms) has substantial impacts on the message-passing scheme that messages between closer nodes are more important. Building on these insights, we propose a principled approach that finds the *radius of influence* for each atom to extract the explanatory subgraphs. These radii of influence define the *localized regions that capture the most important interactions critical for the prediction.*

Overall, we summarize our contribution as follows: ① We are the first to identify and introduce two crucial yet previously overlooked differences between 2D and 3D GNNs. ② Building on this, we propose a principled approach, known as RISE, that leverages radii of influence to extract the explanatory subgraphs for 3D GNNs. ③ Our proposed approach does not require the relaxation from binary masks to continuous masks as in existing methods, allowing exact optimization. ④ Radii of influence naturally account for both differences, making RISE the only explanation pipeline that can produce chemically interpretable explanatory subgraphs as discussed in Sec. 3.3. ⑤ Experiments demonstrate the superiority of our proposed framework, not only in quantitative evaluation metrics but also in producing chemically interpretable results.

# 2. Background and Related Work

## 2.1. GNN Explanation

The notations used throughout the paper are summarized in Table 4 in Appendix A. A GNN $\Phi$ is a mapping from a graph $G$ to a prediction $\hat{Y}$ corresponding to the target

variable $Y$. The target $Y$ can be discrete labels for graph classification tasks or continuous values for regression tasks. Without loss of generality, we focus on regression tasks.

There is a notable absence of well-founded explanation methods specifically designed for 3D GNNs, as most existing approaches work primarily for 2D GNNs. A 2D molecular graph $G$ is represented as $G = (\mathcal{V}, A, X)$, where $\mathcal{V} = \{v_1, v_2, \ldots, v_n\}$ denotes a set of $n$ nodes, $A$ denotes an $n \times n$ adjacency matrix where each entry $a_{ij} \in \{0, 1\}$ indicates the presence or absence of an edge connecting nodes $i$ and $j$, and $X = [\mathbf{x_1}, \mathbf{x_2}, \ldots, \mathbf{x_n}]^T \in \mathbb{R}^{n \times d_v}$ is the node feature matrix, with each $\mathbf{x_i} \in \mathbb{R}^{d_v}$ denoting the $d_v$-dimensional node feature associated with node $v_i$.

Following Ying et al. (2019), instance-level graph explanation aims to find a subgraph $G_S \subseteq G$ that is important to the target $Y$. This is formally expressed as:

$$G_S^* = \arg\min_{G_S \subseteq G} \mathcal{L}(Y; \Phi(G_S)) \quad \text{s.t.} \quad |G_S| \leq B, \quad (1)$$

where $\mathcal{L}$ denotes the task-dependent loss function, and $B$ represents a size constraint on the subgraph to avoid trivial solutions. Eq. (1) can be rewritten as:

$$G_S^* = \arg\min_{M} \mathcal{L}(Y; \Phi(M \odot A, X)) \text{ s.t. } \|M\|_1 \leq B, \quad (2)$$

where $M \in \{0, 1\}^{n \times n}$ are binary masks, indicating whether to retain or remove an edge, applied to extract a subgraph.

## 2.2. Existing Works

GNN explanation methods (Huang et al., 2023; Luo et al., 2020; Ying et al., 2019; Zhang et al., 2024; Yuan et al., 2021; Miao et al., 2023; Xie et al., 2022) aim to understand and interpret the decision-making processes of GNNs. We can classify GNN explanation into two categories—**instance-level** and **model-level** explanations. Instance-level explanation aims to identify important graph components that drive the model's decisions. As a pioneering work, GN-NExplainer (Ying et al., 2019) finds the most important subgraph in a transductive setting by masking the edges. PGExplainer (Luo et al., 2020) extends GNNExplainer to the inductive setting by parametrizing masks with neural networks. Various works have been proposed to further extend and refine these methods. For example, MixupExplainer (Zhang et al., 2023) consider the subgraph distribution to further improve explanatory results. In contrast, model-level methods aim to provide global explanations of a model's behavior, independent of specific input graphs. For instance, XGNN (Yuan et al., 2020) utilizes a graph generator to interpret the model's decision-making process, while CGE (Fang et al., 2023) generates explanatory subgraphs along with their corresponding subnetworks. Despite the advancements in GNN explanation methods and their

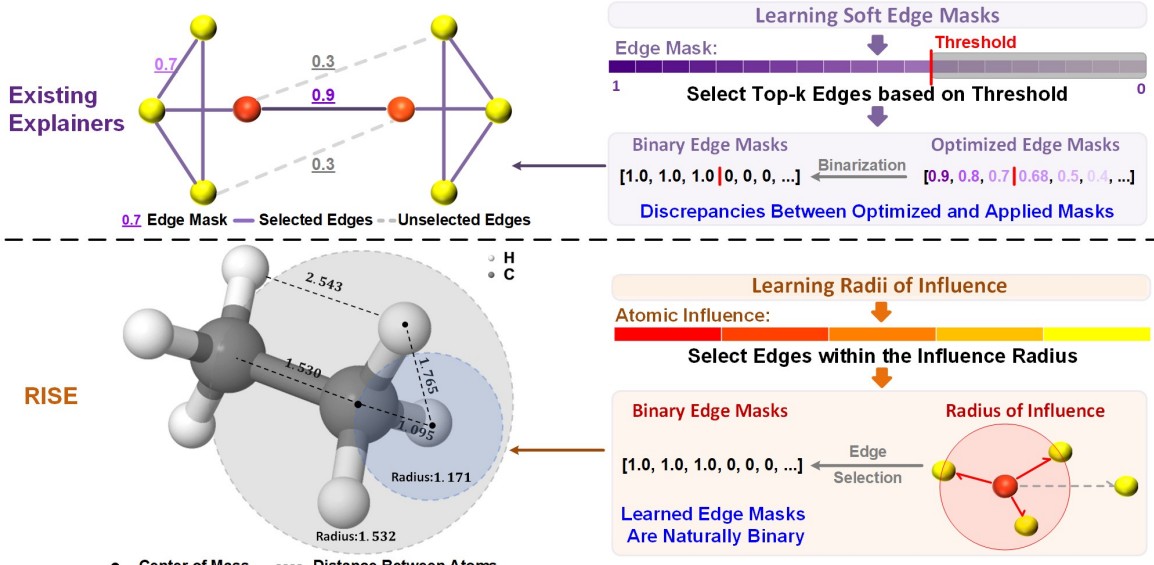

*Figure 1.* Comparison with existing approaches. Existing approaches require a relaxation from binary masks to soft continuous masks, leading to inconsistencies between the optimized masks and the explanatory binary masks. *These inconsistencies not only compromise explanation fidelity quantitatively but also produce chemically uninterpretable results, making the model explanation itself a black-box.* On the other hand, RISE introduces a novel framework for extracting explanatory substructures based on atomic radii of influence that aim to find localized regions that capture the most important interactions decisive to the prediction under some budgets. **Given an appropriate small budget, RISE can precisely extract chemical bonds and chemical bonds only.** For instance, for Ethane ($CH_3CH_3$) in the figure, the radii of influence from our experiments assign the C of interest with a radius of 1.532 and the H of interest with a radius of 1.171. Under similar radii of influence for C atoms and H atoms, respectively, RISE extracts the precise chemical bonds and chemical bonds only: C-H ($1.171 > 1.095$); C-C ($1.532 > 1.530$); all other edges have a distance greater than 1.532 and will be masked out by RISE. Note the unit of distance is $\mathring{A}$, which is used throughout the paper unless otherwise specified.

proven effectiveness, existing approaches struggle to effectively explain 3D GNNs (Miao et al., 2023), as they often overlook the fundamental differences between 2D and 3D representation learning as discussed later in Sec. 3.1. In this work, we focus on addressing the unique challenges of 3D graph representation and learning at the instance level.

**Relations with Prior Works.** There is a significant lack of work in 3D GNN explanation. To our best knowledge, there is only one published work specifically concerned with 3D GNN explanation, known as LRI (Miao et al., 2023). LRI introduces a probabilistic masking framework based on Bernoulli and Gaussian noise to evaluate the importance of atom existence and spatial location, then masking the edges based on the importance of its vertices. *However, it only considers the difference in representation between 2D and 3D molecular graphs; there is an important piece that is missing: The difference in learning mechanism for 2D and 3D GNNs resulting from different representations.* The message-passing framework operates differently with 3D molecular representations. Building on this, we propose a principled approach based on the radius of influence, which accounts for both representational differences and variations in the learning mechanism. This design offers several theoretical advantages and leads to significant performance improvements, as demonstrated in Sec. 3.3 and Sec. 4.2.

## 3. RISE: Radius of Influence based Subgraph Extraction

In this section, we begin by analyzing the two geometric characteristics unique in 3D GNNs in Sec. 3.1: ❶First, we discuss the difference in representation and the challenges it brings to existing explanation methods in Sec. 3.1.1; ❷We then discuss the differences in learning, highlighting the crucial role of proximity in the message-passing scheme in Sec. 3.1.2. Motivated from these geometric characteristics, we introduce our proposed Radius of Influence based Subgraph Extraction (RISE) framework, specifically reformulated for 3D GNNs, in Sec. 3.2. Finally, in Sec. 3.3, we discuss the benefits that RISE offers, especially its consistency in optimization and, most importantly, its capability to provide interpretable explanatory subgraphs. An overview of RISE is given in Fig. 1.

### 3.1. The Missing Piece: Geometric Characteristics

The primary difference between 2D and 3D molecular learning stems from the inherent structural distinctions between the molecular graph representations. In 2D graphs, nodes are presented in a planar layout and the edges are the chemical bonds. In 3D graphs, nodes are presented in 3D Euclidean space, offering a more general depiction of geometric relationships, and the edges are usually constructed

based on a cut-off distance (Schütt et al., 2017; Gasteiger et al., 2020; Liu et al., 2022; Brandstetter et al., 2022), resulting in very dense graphs (difference in representation) as illustrated in Fig. 2. This construction is motivated by the underlying physical and chemical interactions (Wang et al., 2022a; Leach, 2001), aiming to model both bonded and non-bonded interactions (difference in learning). This distinction between 2D and 3D molecular learning leads to two geometric characteristics that should be considered but are missing in existing GNN explanation methods.

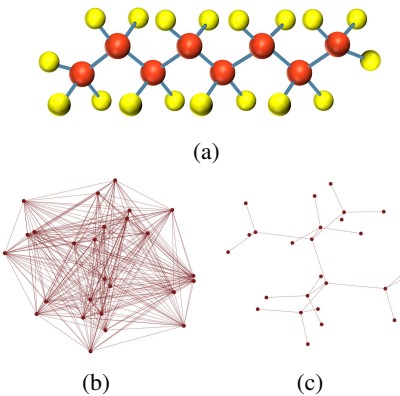

Figure 2. (a): 2D representation of $C_8H_{18}$—Nodes are atoms, and edges are chemical bonds. No geometric information; typically a small number of edges. (b): 3D representation of $C_8H_{18}$—Nodes are atoms with spatial locations. Edges are constructed with a specified cut-off distance, resulting in a dense graph. (c): 3D representation of $C_8H_{18}$ with all non-bonding edges removed.

### 3.1.1. DIFFERENCE IN REPRESENTATION

We find that the difference between 2D and 3D molecular representations poses significant challenges in adapting 2D GNN explanation methods for 3D GNNs. Specifically, to allow gradient-based optimization, the binary hard-masks in Eq. (2) are relaxed to continuous soft-masks, $M_{\text{soft}} \in [0, 1]^{n \times n}$. During testing, an explanatory subgraph is generated from the soft-mask based on the budget; the masks are now made strictly 0-s and 1-s, even though some values, such as $0.70$, might be thresholded to 1, while others, such as $0.68$, are assigned to 0. *Clearly, a significant drawback of relaxing discrete masks to soft masks lies in the difference between optimized continuous masks and the final binary masks representing the explanatory subgraph (inconsistency in optimization).* Mathematically, we give the following bound to illustrate such an inconsistency:

$$
G_S^* = \arg\min_{G_S \subseteq G} \mathcal{L}(Y; \Phi(G_S)) \leq \underbrace{\mathcal{L}(Y; \Phi(X, M_{\text{soft}} \odot A))}_{\text{optimization objective}}
$$
$$
+ \underbrace{\mathcal{L}(\Phi(X, M_{\text{soft}} \odot A); \Phi(X, M \odot A))}_{\text{difference that is ignored during optimization}},
$$
$$
(3)
$$

GNNExplainer (Ying et al., 2019) and follow-up works (Luo et al., 2020; Yuan et al., 2021) attempt to lessen this issue by including additional losses penalizing the extent to which the masks deviate from 0 or 1, such as the entropy loss. Moreover, the budget is enforced through sparsity regularization loss, such as the $L_1$ regularization. Therefore, the final optimization objective becomes

$$
G_S^* = \arg\min_{M_{\text{soft}}} \mathcal{L}(Y; \Phi(M_{\text{soft}} \odot A, X)) + \alpha \|M_{\text{soft}}\|_1
$$
$$
+ \beta \cdot \mathbb{H}[M_{\text{soft}}], \tag{4}
$$

where $\alpha$ and $\beta$ are loss balancing terms. However, adding additional loss terms introduces a trade-off between the optimization objective and the regularization. This trade-off often requires careful tuning of hyperparameters to balance the sparsity or discreteness of the masks against the primary objective of accurately explaining the model's predictions. *This issue exists in 2D GNN explanation as well but worsens in 3D GNNs due to a rapid increase of edges, where small inconsistencies for each mask compound into significant deficits in explanation fidelity.*

### 3.1.2. DIFFERENCE IN LEARNING

The dense edges in 3D GNNs aim to capture interatomic interactions, including both bonded and non-bonded interactions. These interactions and forces usually decay significantly with distance. Interactions between nodes separated by large distances are typically negligible due to the rapid decay of force magnitudes, e.g., London dispersion potential decays as $\frac{1}{d^6}$ or faster, Pauli repulsion decays as $e^{-\alpha d}$, and strong interactions usually have a short-distance dependency ($1 - 2\mathring{A}$ for covalent bonds and $2 - 3\mathring{A}$ for hydrogen bonding).

We find that 3D GNNs can fully leverage the underlying physical principles and truly learn to capture the proximity-dependent nature of interactions. In Sec. 4.1, we conduct extensive experiments to demonstrate that 3D GNNs learn the proximity-dependency in message passing: *Messages between closer nodes in general carry greater importance.* This learning difference between 2D and 3D molecular graphs underscores the need for a tailored explanation framework that effectively captures the unique ability of 3D GNNs to learn range-dependent relationships.

These two geometric characteristics make 2D GNNs poorly suited for 3D GNNs, which can be seen from their poor performance in Sec. 4.2. *We are the first to propose a principled explanation method based on our findings on the geometric characteristics of 3D molecular graphs. The geometric influence on the message-passing scheme suggests reformulating 3D graph explanations in terms of atomic radii of influence, explicitly accounting for the varying importance of interactions based on distance.*

## 3.2. RISE Reformulation

Due to the difference in representation, where 3D molecular graphs are constructed based on a cut-off distance, 3D graphs can be treated as *proximity graphs*, in which edges are not explicitly defined.

**Definition 3.1.** A *directed proximity graph* (DPG) is a geometric graph constructed from a set of points in a metric space, where directed edges are formed based on a proximity rule. Specifically, each node $v_i$ has an associated radius $0 \leq r_i$, and a directed edge $e_{i \mapsto j}$ exists if and only if $d_{ij} < r_i$, where $d_{ij}$ is the distance between $v_i$ and $v_j$.

*Remark* 3.2. Any 3D graph constructed based on node radii can be viewed as a directed proximity graph. A 3D graph constructed based on a cut-off distance (the same radius for all nodes) is a proximity graph under uniform radii; the graph is undirected as all the radii are the same.

In 2D GNNs, edges are explicitly defined, leading to the formulation of 2D graphs as described in Sec. 2.1, where subgraphs are considered subsets of these edges. In contrast, 3D GNNs do not have explicitly defined edges; instead, edges are determined based on a specific cut-off radius. Building on this concept, we generalize graph construction using the notion of DPGs; consequently, explanatory subgraphs are also identified as DPGs as presented below.

We define a 3D graph as $G = (\mathcal{V}, P, R, X)$, where $\mathcal{V}$ represents the set of nodes and $X$ denotes their features, similar to 2D GNNs. The matrix $P \in \mathbb{R}^{n \times 3}$ contains the Euclidean coordinates of the nodes, and $R = [r_1, r_2, \ldots, r_n] \in \mathbb{R}^n$ specifies their radii for edge constructions.

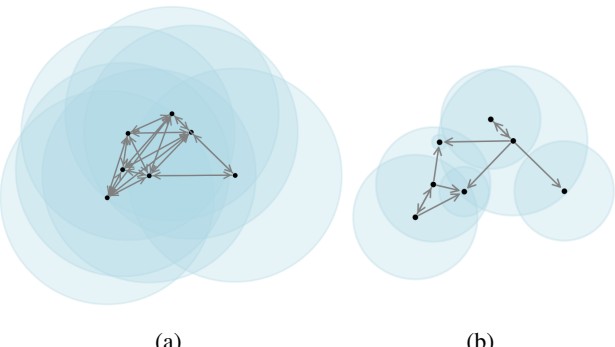

(a)                  (b)

*Figure 3.* (a): Original 3D graphs constructed based on a common cut-off distance; this is the approach taken in most 3D GNNs (Schütt et al., 2017; Gasteiger et al., 2020; Liu et al., 2022; Brandstetter et al., 2022). The edges are bidirectional and dense. (b): Explanatory substructure identified by finding the radii of influence. The radii of influence are optimized (the circles **shrink** dynamically; see an illustration in Fig. 6 in Appendix B) such that the critical messages that are most relevant to the prediction task are preserved. The edges are directed and more sparse.

With this new definition of 3D graphs, instance-level graph

explanation in Eq. (1) can be reformulated as:

$$G_S^* = \arg\min_{\mathbf{M^r}} \mathcal{L}(Y; \Phi(P, M^r \odot R, X)) \text{ s.t. } \|M^r\|_1 \leq B, \tag{5}$$

where $M^r \in [0,1]^n$ are continuous masks defining the *radius of influence* of each node as shown in Fig. 3. $B$ is the budget to avoid trivial solutions and can be set as a ratio $B = \rho \cdot \|R\|_1$.

Clearly, the resulting explanatory graph is a DPG and is a subgraph of the original graph. The main objective of 3D GNN explanation is to provide interpretable insights into the predictions. The concept of the *radius of influence* inherently contributes to chemical explainability, as it represents the spatial extent over which each atom interacts with others, depending on the property of consideration. Different chemical properties are influenced by different types of atomic interactions, which may only be significant within specific spatial ranges.

Let $A^r \in \{0,1\}^{n \times n}$ be the adjacency matrix induced by the radii, $R$, of the original graph. Eq. (5) can be optimized by masking of the computation graph of $A^r$ through:

$$G_S^* = \arg\min_{\mathbf{M}} \mathcal{L}(Y; \Phi(P, M \odot A^r, X)), \tag{6}$$

where $M \in \{0,1\}^{n \times n}$ indicates the removal or retention of an edge after reducing the radii of influence, $M^r$. As strict discrete values are not differentiable, we reparametrize each element $M_{ij}$ in $M$ with a bounded differentiable function $f$ such that $M_{ij} = f(M_i^r, d_{ij})$ that satisfies $M_{ij} \to 1$ when $d_{ij} < M_i^r$ and $M_{ij} \to 0$ when $d_{ij} > M_i^r$, where $d_{ij}$ is the distance between nodes $v_i$ and $v_j$, and $M_i^r$ denotes the radius of influence of the node $v_i$. Generally, any bounded differentiable function that satisfies this requirement should suffice. However, a smooth monotonic function is better for the purpose of gradient-based optimization. Therefore, we use the following function in particular:

$$M_{ij} = \frac{1}{1 + e^{-k(M_i^r - d_{ij})}}, \tag{7}$$

where $k > 0$ is typically large. With a sufficiently large $k$, it is obvious that the requirement above is satisfied.

### 3.3. Merits of RISE

**Consistency in Optimization.** RISE is consistent between the optimization objective and the explanation process, as it directly optimizes the radius of influence, which is naturally continuous; there is no relaxation from discrete masks to continuous masks. The optimized radius mask $M^r$ naturally represent a subgraph $G_s = (\mathcal{V}, P, M^r \cdot R, X)$ whereas in existing methods, the optimized continuous soft edge or node masks do not represent a subgraph $G_s \neq (\mathcal{V}, M_{\text{soft}} \odot A^r, X)$. Additionally, we can enforce the budget exactly to

avoid having a sparsity regularization loss:

$$M_i^r = B \cdot \frac{\exp(\Theta_i)}{\sum_{j=1}^n \exp(\Theta_j)} \cdot \sigma(\Omega_i), \qquad (8)$$

where $\Theta, \Omega \in \mathbb{R}^n$ are both learnable parameters and $\sigma(\cdot)$ is the sigmoid function. It is ensured by construction that $|M^r|_1 \leq B$. Therefore, RISE is consistent in optimization, as it does not require converting discrete values into continuous ones for optimization and does not include penalty or regularization terms to promote discreteness or enforce the budget, different from existing works where the inconsistency is induced as seen in Eq. (4). *Such inconsistency not only compromises explanation fidelity but also presents a far more severe issue: The explanatory substructures become chemically uninterpretable, making model explanation itself a black-box.* We demonstrate this below.

**Interpretable Subgraph Extraction.** In 2D GNNs, edges correspond directly to chemical bonds, naturally forming the basis for chemically interpretable substructures. In 3D GNNs, due to a rapid increase of edges, different continuous edge masks can all yield very high explanation fidelity, leading to ambiguity in explanation. On the other hand, enforcing discreteness regularization loss can compromise explanation fidelity, ultimately reducing the reliability of the extracted substructures. LRI shifts edge masks to node masks by defining each edge mask as the product of the node masks of its end nodes. However, this approach still fails to provide chemically interpretable substructures: two nodes with high mask values will indicate a high importance for their connecting edge, even if they are distant (see more details in Appendix C). This misrepresentation contradicts both the learning dynamics of 3D GNNs and actual chemistry.

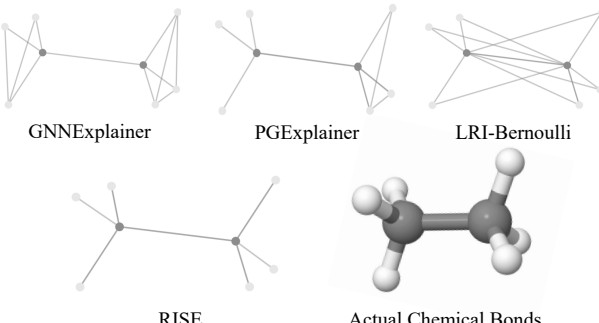

GNNExplainer       PGExplainer       LRI-Bernoulli

RISE              Actual Chemical Bonds

*Figure 4.* Explanatory substructure produced from experiments by different explanation methods on the Ethane molecule ($CH_3CH_3$) in the QM9 dataset (Ramakrishnan et al., 2014). **The same budget (number of edges) is used for different explainers.** It is obvious that only RISE yields chemically interpretable results that conform to interpretable chemical structures. It should be noted that, under larger budgets, baseline methods yield a more "chaotic" set of edges that are uninterpretable at all, whereas RISE identifies atomic regions of influence, enabling chemical interpretation. This underscores the need for interpretable 3D methods like RISE.

With appropriate radii of influence under a small budget, RISE is able to extract chemical bonds and chemical bonds only as illustrated in Fig. 1, while all other methods may fail due to the aforementioned issues. In Fig. 4, we show the explanatory substructure from real experiments; it is obvious that all baseline methods extract chemically uninterpretable results while RISE extracts exactly the chemical bonds. We provide more results in Fig. 8 in Appendix D. Therefore, designing interpretable explanation methods like RISE for 3D GNNs is critical.

## 4. Experiments

In this section, we begin by addressing the critical question of how the distances between nodes influence the importance of message passing; as discussed in Sec. 4.1, it is clear that the distances indicate the importance of message passing. Next, we evaluate our RISE method, developed based on insights from the aforementioned findings, and compare its performance against several state-of-the-art (SOTA) baselines in Sec. 4.2. Several representative 3D GNNs, including invariant models SchNet (Schütt et al., 2017) and DimeNet (Gasteiger et al., 2020) and an equivariant model SEGNN (Brandstetter et al., 2022), are used as backbone models. An introduction to these 3D GNNs is given in Appendix E. All explanation methods are evaluated on the widely used datasets QM9 (molecules in the equilibrium state) (Ramakrishnan et al., 2014) and GEOM (geometric ensemble of molecules) (Axelrod & Gomez-Bombarelli, 2022). The results highlight the superiority of RISE as a principled approach specifically designed for 3D GNNs.

### 4.1. Proximity Decides Messages To Pass or Not To Pass

To study where proximity (distance) decides the importance of message passing, we conduct experiments centered on two key questions: ❶ Does excluding short-distance edges from message passing impact the model's predictions more significantly than excluding long-distance edges? ❷ Do edges with similar distances have comparable importance in influencing the model's predictions?

**Setup.** We group edges based on their distances $d_{ij}$ into 5 annular bins: $\{0 \leq d_k \leq d_{ij} < d_{k+1} \leq d_{\text{cut-off}}\}_{k=1}^5$, where all $d_k$-values are determined such that, on average over the entire testing dataset, each annulus contains $\sim 20\%$ of the total edges. ❶ To answer the first question, we remove all the edges in each annulus and measure the resulting decline in predictive fidelity. If the importance of message-passing edges correlates with distance, the removal of edges from inner annuli (shorter distances) should lead to a more pronounced decline in performance. ❷ To answer the second question, we randomly remove $10\%$ of the edges within each annulus and evaluate the model performance after the removal of edges; if the intra-annulus variance is

*Table 1.* Comparison of masking edges in different radius ranges on $\alpha$ and $\epsilon_{\text{LUMO}}$ properties of the QM9 dataset with SchNet and SEGNN as the representative of invariant and equivariant models, respectively. Values in most cells are strictly greater than the previous ones in the row; clearly, proximity decides the importance of message passing.

| Property | Backbone | Original MAE | MAE After Annulus of Edges Removed | | | | |
|---|---|---|---|---|---|---|---|
| | | | 80-100% | 60-80% | 40-60% | 20-40% | 0-20% |
| $\alpha$ | SchNet | 0.118 | 0.248 | 0.253 | 0.373 | 0.373 | 0.739 |
| | SEGNN | 0.110 | 0.196 | 0.342 | 0.418 | 0.826 | 0.895 |
| $\epsilon_{\text{HOMO}}$ | SchNet | 0.046 | 0.051 | 0.051 | 0.058 | 0.070 | 0.128 |
| | SEGNN | 0.032 | 0.034 | 0.035 | 0.036 | 0.048 | 0.088 |

consistently small, it suggests that the importance of edges in each annulus (similar distance) is approximately uniform.

**Results.** The quantitative results for setting ❶ are presented in Table 1. It is evident that removing closer annuli leads to a more significant drop in MAE, as values in most cells strictly decrease compared to the previous row. A sample of the qualitative results for setting ❷ is shown in Fig. 5, with the full results, all sharing the same trend, available in Appendix F. From these results, we observe that edges at similar distances have comparable importance, as indicated by the relatively flat trend lines with minimal fluctuation. Additionally, shorter edges play a more significant role in message passing. In summary, proximity determines the importance of messages. This pattern holds consistently across molecular graphs in the dataset, reinforcing the general trend. For each molecular graph, we identify the atomic radii of influence to retain the most critical interactions within the given budget.

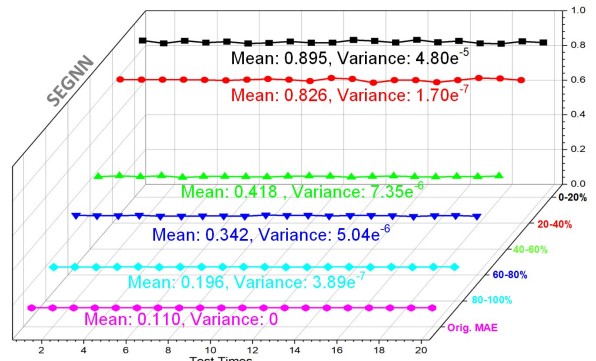

*Figure 5.* The visualization of the quantitative results of $\alpha$ of the QM9 dataset on SEGNN when randomly masking 10% edges in different annuli. It shows that the influence of edge masking has a significant correlation with the distances, i.e., masking short edges will cause larger perturbation than long edges. Moreover, the small variance of each annulus demonstrates the comparable importance among edges with similar distances.

### 4.2. The Power of Proximity: RISE Performance

We evaluate the performance of RISE on two widely used molecular datasets—QM9 and GEOM. For the QM9 dataset, the learning objective is a regression task to predict molecular properties for stable molecules; we perform experiments on 4 important properties: the dipole moment ($\mu$), isotropic polarizability ($\alpha$), the highest occupied molecular orbital energy ($\epsilon_{\text{HOMO}}$), and the lowest unoccupied molecular orbital energy ($\epsilon_{\text{LUMO}}$). For the GEOM dataset, the learning task focuses on predicting the energy of conformers at their low-energy states.

**Setup.** Experiments are conducted on all three representative 3D GNNs as the backbone models. Specifically, invariant models are tested with the original QM9 test dataset and $1,000$ randomly extracted molecules from the GEOM dataset, while the equivariant model is tested using the first $1,000$ molecules from the QM9 test dataset and the same $1,000$ molecules from the GEOM dataset due to the computational hurdle of tensor decompositions in SEGNN. RISE is compared with the two representative explanation methods, GNNExplainer and PGExplainer, and the only existing 3D explanation method—LRI-Bernoulli. Note there is another method in LRI, LRI-Gaussian; however, it is perturbation-based and cannot be directly compared with RISE. More details about baseline explanation methods are in Appendix G. Following existing works (Yuan et al., 2022), we report the mean absolute error (MAE, the lower the better) of the predictions made using the explanatory subgraphs.

To thoroughly evaluate performance across varying levels of edge sparsity, we set the budget parameter $\rho$ to different values: $0.3$ to $0.6$ for the QM9 dataset, and $0.4$ to $0.6$ for the GEOM dataset. Since different explanation methods have different budget constraints, we compare them in a unified way based on the number of preserved edges. ***The baseline models' budgets are set such that they strictly preserve more edges than RISE, making the comparison even more advantageous for the baselines.*** More experimental details including hyperparameter settings and training details are attached in Appendix H.

**Results.** We present the results on the QM9 dataset using SchNet as a representative of invariant backbone models in Table 2. When SchNet is used as the backbone model, RISE consistently outperforms all baselines across various budgets, except for the $\epsilon_{\text{HOMO}}$ property, where it performs on par with GNNExplainer. We present the results on the QM9 dataset using SEGNN as a representative equivariant backbone in Table 3. Similarly, when SEGNN is used as the backbone model, RISE consistently outperforms all baselines across various budgets, except for the $\alpha$ property

*Table 2.* Resulting MAE (the lower the better) with **SchNet** on the QM9 dataset. The original SchNet employs a cut-off of $10\mathring{A}$, leading to an average of 316 edges per graph. The average number of edges preserved is indicated in parentheses. The best and second best results are respectively highlighted in red and blue, or both in red if they are very close and the lower one has more edges preserved.

| Explainer | $\mu$ | | | $\alpha$ | | |
| --- | --- | --- | --- | --- | --- | --- |
| | 0.3 | 0.4 | 0.5 | 0.3 | 0.4 | 0.5 |
| GNNExplainer | 2.972(50.0%) | **1.162(69.9%)** | **0.859(79.9%)** | **2.836(50.0%)** | **2.216(69.9%)** | **2.054(79.9%)** |
| PGExplainer | **1.884(45.6%)** | 1.756(65.1%) | 1.684(82.8%) | 10.420(45.9%) | 9.520(63.9%) | 9.221(81.9%) |
| LRI-Bernoulli | 6.087(44.1%) | 5.325(70.0%) | 4.127(80.0%) | 14.459(47.8%) | 11.847(67.3%) | 10.144(80.3%) |
| **RISE (ours)** | **0.596(44.1%)** | **0.518(64.8%)** | **0.422(79.2%)** | **2.670(44.6%)** | **2.127(63.7%)** | **1.430(77.0%)** |
| Explainer | $\epsilon_{\text{HOMO}}$ | | | $\epsilon_{\text{LUMO}}$ | | |
| | 0.3 | 0.4 | 0.5 | 0.3 | 0.4 | 0.5 |
| GNNExplainer | **0.334(49.9%)** | **0.239(69.9%)** | **0.154(80.0%)** | **1.181(50.0%)** | **0.766(69.9%)** | **0.209(89.9%)** |
| PGExplainer | 0.567(50.0%) | 0.580(70.0%) | 0.598(80.0%) | 1.322(50.2%) | 1.355(71.6%) | 1.418(85.9%) |
| LRI-Bernoulli | 0.601(47.7%) | 0.622(67.4%) | 0.623(80.4%) | 1.355(50.0%) | 1.379(80.3%) | 1.401(90.0%) |
| **RISE (ours)** | **0.333(44.3%)** | **0.248(63.5%)** | **0.172(76.8%)** | **0.544(45.2%)** | **0.370(66.0%)** | **0.207(80.4%)** |

*Table 3.* Resulting MAE (the lower the better) with **SEGNN** on the QM9 dataset. The original work employs a cut-off of $5\mathring{A}$, leading to an average of 280 edges per graph. The average number of edges preserved is indicated in parentheses. The best and second best results are respectively highlighted in red and blue, or both in red if they are very close and the lower one has more edges preserved.

| Explainer | $\mu$ | | | $\alpha$ | | |
| --- | --- | --- | --- | --- | --- | --- |
| | 0.4 | 0.5 | 0.6 | 0.4 | 0.5 | 0.6 |
| GNNExplainer | 1.183(29.8%) | 1.056(50.0%) | 0.939(59.8%) | 27.590(29.8%) | 23.026(44.9%) | 16.388(59.8%) |
| PGExplainer | 1.197(30.1%) | 1.140(40.1%) | 0.974(60.2%) | 27.191(31.5%) | 23.320(41.5%) | 17.673(52.2%) |
| LRI-Bernoulli | **1.051(42.1%)** | **1.005(50.0%)** | **0.895(60.0%)** | **26.247(30.4%)** | **22.625(43.2%)** | **15.201(60.2%)** |
| **RISE (ours)** | **1.040(27.2%)** | **0.872(40.0%)** | **0.674(51.8%)** | **26.261(27.9%)** | **20.274(40.2%)** | **15.557(51.6%)** |
| Explainer | $\epsilon_{\text{HOMO}}$ | | | $\epsilon_{\text{LUMO}}$ | | |
| | 0.4 | 0.5 | 0.6 | 0.4 | 0.5 | 0.6 |
| GNNExplainer | 0.849(29.8%) | **0.725(39.8%)** | **0.457(59.8%)** | 1.671(29.8%) | 1.455(39.8%) | 1.014(59.8%) |
| PGExplainer | **0.848(29.8%)** | 0.766(40.0%) | 0.461(60.0%) | 1.778(30.6%) | 1.582(40.6%) | 1.129(60.5%) |
| LRI-Bernoulli | 3.677(30.0%) | 3.646(40.0%) | 3.363(60.0%) | **1.006(30.0%)** | **0.724(40.0%)** | **0.578(63.8%)** |
| **RISE (ours)** | **0.838(26.8%)** | **0.551(39.4%)** | **0.339(50.9%)** | **0.988(26.8%)** | **0.679(39.1%)** | **0.438(50.5%)** |

under a budget of $0.4$, where it performs on par with LRI-Bernoulli. When using SEGNN as the backbone model, higher MAEs are observed across the board.

Additionally, in Appendix I, we present results on the QM9 dataset using DimeNet (invariant) and on the GEOM dataset. RISE consistently achieves the best performance across all configurations on the GEOM dataset and outperforms other methods in most configurations on QM9 with DimeNet. These results further emphasize RISE's ability to capture meaningful interactions essential for molecular prediction while generating interpretable explanatory substructures that generalize across different datasets and tasks.

While achieving exceptional quantitative results is important, another crucial aspect is the interpretability of the explanation results. As discussed in Sec. 3.1, due to the differences between 2D and 3D GNNs in molecular representation and learning dynamics, existing methods often produce chemically uninterpretable results when applied to 3D GNNs. We provide several visualizations of the explanatory substructures in Appendix D. Notably, under a small budget, when explanation methods can preserve only a limited number of edges, RISE is the only method that selectively retains edges corresponding exclusively to chemical bonds. This is particularly significant because, chemically,

bonds represent the strongest interactions, and RISE effectively captures this crucial aspect of molecular structures.

## 5. Conclusions, Limitations, and Future Work

In this work, we introduce a principled explanation method for 3D GNNs by representing 3D graphs as directed proximity graphs and determining the radius of influence for each atom. Our approach is the first to explicitly address the fundamental distinctions between 2D and 3D GNNs in both representation and learning dynamics. Consequently, it surpasses existing SOTA explainers in quantitative performance on 3D GNNs. More importantly, existing methods often yield chemically uninterpretable explanations for 3D GNNs, as they do not specifically account for 3D characteristics, rendering the explanation method itself a black box. In contrast, our method consistently produces interpretable and chemically meaningful results.

**Limitations and Future work.** This work focuses on explaining relatively small 3D molecular graphs. However, macromolecules, such as proteins, exhibit more complex interactions and intricate structural hierarchies that may introduce long-range dependencies. It would be interesting to see if our method can be generalized to macromolecules with long-range interactions.

## Acknowledgments

The work was partially supported by National Science Foundation award #2421839 and institutional funds of Stony Brook University.

## Impact Statement

This paper advances the field of 3D Graph Neural Networks by introducing a novel framework that enhances 3D GNN explanation through localized influence radii, capturing spatial and structural interactions critical for predictions. Our method aligns with the geometric and physical properties of 3D data, supporting trustworthy AI applications in scientific domains like molecular discovery. There are limited potential societal consequences of our work, and none require specific highlighting at this time.

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

## A. Table of Notations

*Table 4.* A summary of used notations.

| Symbol | Description |
|--------|-------------|
| $\Phi$ | A GNN mapping a graph $G$ to a prediction $\hat{Y}$. |
| $G$ | A molecular graph, defined as $G = (\mathcal{V}, A, X)$ for 2D and $G = (\mathcal{V}, P, R, X)$ for 3D. |
| $\mathcal{V}$ | Set of nodes in the graph, $\mathcal{V} = \{v_1, v_2, \ldots, v_n\}$. |
| $A$ | Adjacency matrix of size $n \times n$, where $a_{ij} \in \{0, 1\}$ indicates edge presence. |
| $X$ | Node feature matrix, $X \in \mathbb{R}^{n \times d_v}$, where $\mathbf{x_i} \in \mathbb{R}^{d_v}$ is the feature of node $v_i$. |
| $P$ | Euclidean coordinates of nodes in 3D space, $P \in \mathbb{R}^{n \times 3}$. |
| $R$ | Initial radii for edge construction in 3D graphs, $R = [r_1, r_2, \ldots, r_n] \in \mathbb{R}^n$. |
| $\alpha, \beta$ | Loss balancing terms. |
| $\mathcal{L}$ | Task-dependent loss function. |
| $d_{ij}$ | Euclidean distance between nodes $v_i$ and $v_j$. |
| $G_S$ | A subgraph of $G$. |
| $M$ | Binary (hard) edge mask, $M \in \{0, 1\}^{n \times n}$, used for subgraph extraction. |
| $M_{\text{soft}}$ | Continuous (soft) edge mask, $M_{\text{soft}} \in [0, 1]^{n \times n}$ for approximation of $M$ for gradient-based optimization |
| $M^r$ | Radius of influence masks for nodes in 3D graphs, $M^r \in [0, 1]^n$. |
| $A^r$ | Adjacency matrix induced by radii $R$ in 3D graphs. |
| $\Theta, \Omega$ | Learnable parameters for determining radii of influence in RISE. |
| $\sigma(\cdot)$ | Sigmoid activation function. |
| $B$ | Budget constraint on subgraph size, $B = \rho \cdot \|R\|_1$ with the ratio $\rho$. |
| $\mathbb{H}$ | Entropy function used in soft mask regularization. |

## B. An Animation of Optimizing the Radii of Influence

As we optimize the masks for the radii of influence, we are essentially shrinking the atomic influence circles. An illustration is given in Fig. 6. From the top-left to the bottom-right, the sequence represents the entire evolution.

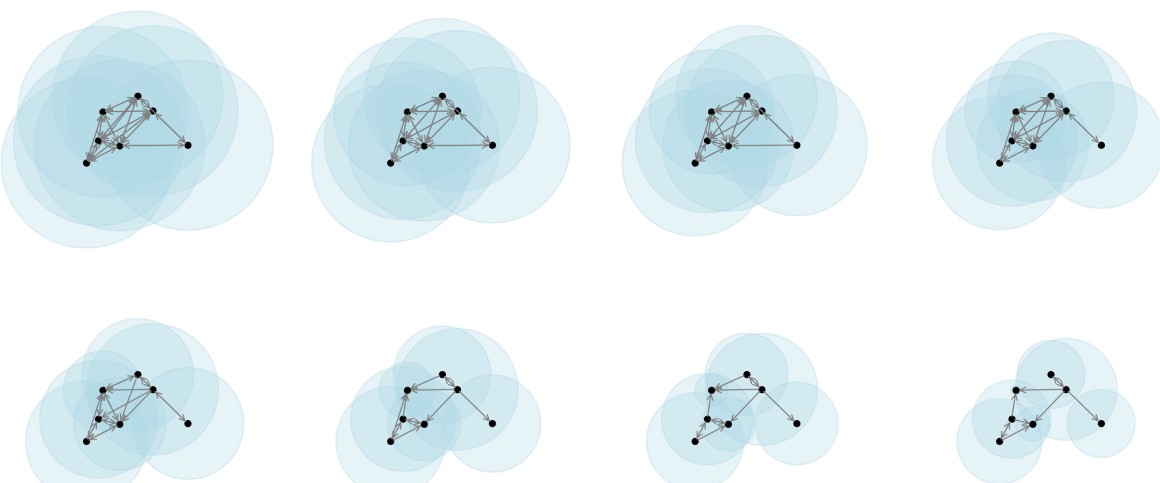

*Figure 6.* An animation of optimizing the radii of influence.

## C. Issues with Masking Nodes in 3D GNNs

The structural differences between 2D and 3D molecular graphs not only alter the representation but also impact the learning mechanism. The importance of message passing should be based on distances. LRI-Bernoulli accounts for the difference in representation by offsetting the edge masks to node masks. Namely, let $M_{\text{node}}$ represent the set of masks on nodes, with $M_{node}^i$ indicating the mask on node $v_i$. LRI-Bernoulli conditions on the message passing by converting node masks to edge masks through:

$$M_{\text{soft}}^{ij} = M_{node}^i \cdot M_{node}^j. \tag{9}$$

This node masking approach, which disregards the differences in learning dynamics between 2D and 3D GNNs, may result in significant issues. We now assume fully connected graphs for simplicity; in fact, with a cut-off distance of 5 , many 3D graphs for small molecules are fully connected (Satorras et al., 2021). Taking $CH_3CH_3$ as an example, both C-C and same-side C-H bonds are critical, so we aim to assign high mask values to the corresponding C and H atoms. However, since all C and H atoms participate in C-C and same-side C-H bonds, assigning high mask values to all C and H atoms fails to differentiate the relative importance of specific bonds. This approach also inadvertently assigns high mask values to less significant interactions, such as H-H edges across different sides. **On the other hand, by accounting for the differences in learning dynamics, the RISE explanation, with appropriate radii of influence, precisely preserves the chemical bonds and chemical bonds only. This is demonstrated in Fig. 7.**

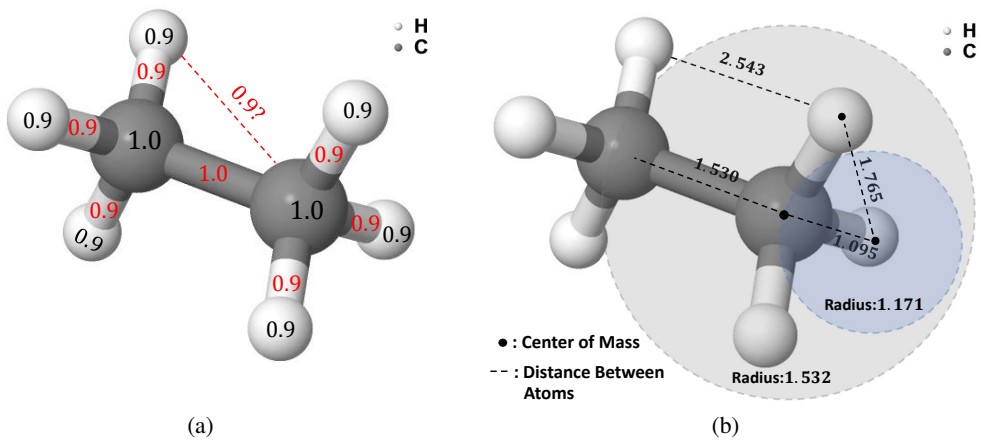

(a)                                    (b)

*Figure 7.* Comparison between node-masking methods (a) and RISE (b). (a): Node-masking methods fail to accurately capture important interactions because they do not consider spatial proximity. Two nodes with high mask values will always result in a high importance for their connecting edge, even if they are far apart. This contradicts how 3D GNNs learn and the principles of actual chemistry. (b): RISE can correctly capture the most important interactions by considering proximity. **Given an appropriate small budget, in this example, $B = 0.15$, RISE can precisely extract chemical bonds and chemical bonds only**, i.e., for Ethane ($CH_3CH_3$) given in the figure, the radii of influence from our experiments assign the C of interest with a radius of 1.532 and the H of interest with a radius of 1.171. Under similar radii of influence for C atoms and H atoms, respectively, RISE extracts precisely chemical bonds and chemical bonds only: C-H ($1.171 > 1.095$); C-C ($1.532 > 1.530$); all other edges have a distance greater than $1.532$ and will be masked out by RISE.

## D. Visualization of Explanatory Substructures

The qualitative results of explanations are depicted in Fig. 8. These results highlight the unique advantage of RISE not only in delivering the highest explanation fidelity (quantitative metric) but also in extracting interpretable subgraphs that are chemically meaningful. Existing explanation methods, including GNNExplainer, PGExplainer, and LRI, often produce fragmented, overly dense, and chemically inconsistent explanations, and this issue gets worse in larger graphs (for visualization purposes, we provide relatively small molecules here).

For instance, in the case of methylene oxalate ($C_3H_2O_4$), other methods fail to isolate all C–O bonds, instead erroneously highlighting distant or non-adjacent nodes due to their reliance on continuous or indirect edge attributions. In contrast, RISE successfully reconstructs the precise chemical structures, as confirmed by structural comparisons with the ground-truth molecular bonds.

The failure of baseline methods stems from their inability to resolve the ambiguity of 3D explanations—either by producing overly distributed importance scores (e.g., LRI) or by failing to enforce discrete structural preservation (e.g., GNNExplainer) as discussed in Sec. 3.3. These limitations underscore the necessity of RISE, a principled approach that resolves the aforementioned issues by considering the two major differences between 2D and 3D graphs.

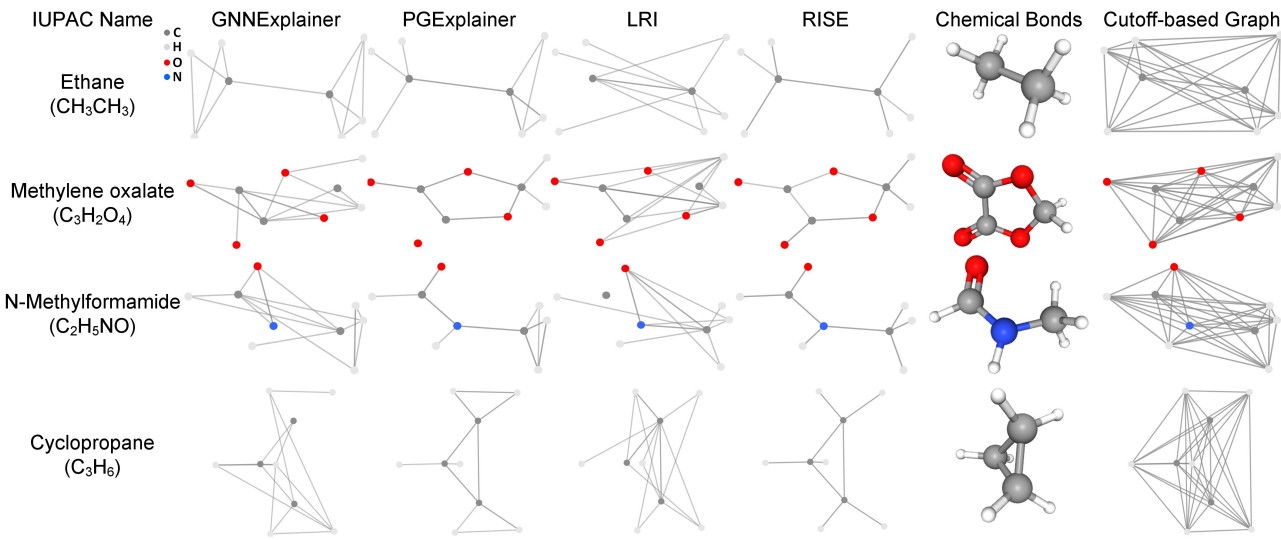

*Figure 8.* The visualized samples of molecules within the QM9 dataset. The explanatory results of GNNExplainer, PGExplainer, LRI-Bernoulli, and RISE are inferred based on SchNet. The budgets are the same across different explanation methods; in other words, the same number of edges are retained in the explanation results. All edges are directed due to masking; however, RISE consistently identifies both directions, as interactions should be bidirectional in nature, whereas other methods often capture only one direction, leading to a less faithful representation of the underlying interactions. As a result, other explanation methods appear to retain more edges visually (both bidirectional and unidirectional edges are visualized as a single edge). Apparently, it can be seen that only our method preserves the chemical structures well, while other methods produce ambiguous explanations without interpretable chemical structures.

## E. 3D Geometric GNNs: Invariant and Equivariant GNNs

In 3D GNNs, molecular or spatial structures are typically represented as 3D graphs, where nodes correspond to atoms or spatial points, and edges indicate interactions or proximity relationships. The most common approach to constructing a 3D graph is to connect nodes based on a distance cut-off, where an edge is established between nodes whose Euclidean distance falls below a specified threshold, i.e., the cut-off. Based on different geometric feature representations of the constructed graph, the corresponding 3D GNNs can be divided into invariant and equivariant GNNs.

Invariant GNNs, also referred to as scalarization GNNs (Duval et al., 2024), are 3D geometric graph neural networks that rely exclusively on invariant features, such as distances, angles, and bond angles. These features remain unchanged under Euclidean transformations, including translations and rotations. As a result, these GNNs are inherently invariant to Euclidean transformations, making them particularly suitable for invariant tasks, such as predicting energies, $\epsilon_{HOMO}$-$\epsilon_{LUMO}$ gaps, and other scalar molecular properties. Notable, but far from complete, examples of invariant GNNs include SchNet (Schütt et al., 2017), DimeNet (Gasteiger et al., 2020), and SphereNet (Liu et al., 2022).

Equivariant GNNs, on the other hand, are designed to handle vectorial or tensorial properties that transform predictably under Euclidean transformations. These networks incorporate both invariant and equivariant features to ensure that the outputs transform in a consistent manner when the inputs are subjected to translations or rotations. For example, when predicting forces, dipole moments, or gradients, the outputs of an equivariant GNN should rotate or translate in accordance with the same transformation applied to the input geometry. Equivariant GNNs leverage advanced mechanisms, such as spherical harmonics and tensor algebra, to ensure that all the operations preserve equivariance throughout the neural network. Notable examples of equivariant GNNs include SE(3)-Transformer (Fuchs et al., 2020), E(3)-equivariant GNNs (Batzner et al., 2022), and SEGNN (Brandstetter et al., 2022).

## F. Proximity Decides Messages To Pass or Not To Pass

Table 5. Corresponding cut-off distance range of annuli.

| Cut-off Distance | 0-20% | 20-40% | 40-60% | 60-80% | 80-100% |
|---|---|---|---|---|---|
| $5\mathring{A}$ | $[0, 2.1]$ | $[2.1, 2.6]$ | $[2.6, 3.2]$ | $[3.2, 4.0]$ | $[4.0, 5.0]$ |
| $10\mathring{A}$ | $[0, 2.2]$ | $[2.2, 2.8]$ | $[2.8, 3.5]$ | $[3.5, 4.4]$ | $[4.4, 10]$ |

To investigate the correlation between edge distance and importance, we conducted extensive experiments on the QM9 dataset using both invariant and equivariant backbones. First, to categorize edges into distinct annuli, we analyzed the edge distance distribution within QM9, as presented in Table 5. The visual and quantitative results are illustrated in Fig. 9.

Our findings reveal that as the distance of masked edges increases, the perturbation induced by edge masking correspondingly decreases, indicating a reduction in edge importance. Additionally, the low variance observed across 20 trials suggests that edges within the same annulus exhibit similar importance. Collectively, these results establish a strong correlation between edge proximity and its role in message passing, reinforcing the significance of RISE, which explains 3D graphs based on spatially localized interactions.

## G. Descriptions of Baseline Explanation Methods

Graph explanation methods aim to find a subgraph that best preserves the predictive signal of the original graph $G$ while satisfying a budget constraint $B$ on the size of the subgraph (Ying et al., 2019). Mathematically,

$$G_S^* = \arg\min_{G_S \subseteq G} \mathcal{L}(Y; \Phi(G_S)) \quad \text{s.t.} \quad |G_S| \leq B, \tag{10}$$

where $\mathcal{L}$ denotes the task-dependent loss function, and $B$ represents a size constraint on the subgraph to avoid trivial solutions. Eq. (10) can be rewritten as:

$$G_S^* = \arg\min_{M} \mathcal{L}(Y; \Phi(M \odot A, X)) \text{ s.t. } \|M\|_1 \leq B, \tag{11}$$

where $M \in \{0, 1\}^{n \times n}$ are binary masks, indicating whether to retain or remove an edge, applied to extract a subgraph.

Existing explanation methods relax binary masks into continuous soft masks, $M_{\text{soft}} \in [0, 1]^{n \times n}$, to enable gradient-based optimization. In practice, GNNExplainer (Ying et al., 2019) initializes these masks as learnable parameters, applying a sigmoid function to constrain their values to be between $0$ and $1$, and optimizes them in a transductive setting.

PGExplainer (Luo et al., 2020) extends this idea by introducing a parameterized explainer that employs a multi-layer perceptron (MLP) to generate edge masks from learned feature embeddings. This approach enables the collective explanation of multiple instances, improving generalization and making it applicable in the inductive setting.

Learnable Randomness Injection (LRI) (Miao et al., 2023) introduces two different methods—LRI-Bernoulli and LRI-Gaussian. LRI-Bernoulli injects Bernoulli noise into nodes to evaluate the importance of node-wise existence for the final prediction; in simple terms, it applies binary masks to nodes. LRI-Gaussian, on the other hand, is a perturbation-based method that adds Gaussian noise to the positions of nodes to assess the significance of their geometric features. Unlike methods that aim to identify substructures, LRI-Gaussian does not identify subgraph but rather focuses on identifying important geometric aspects of the input geometric graph. Therefore, it is not directly comparable with RISE or other extraction-based baselines, including GNNExplainer and PGExplainer.

## H. Experimental Details

We conduct comparative experiments on molecular graph regression tasks using two widely used molecular datasets, QM9 and GEOM. For the QM9 dataset, four key molecular properties are taken for testing purposes: dipole moment ($\mu$), isotropic polarizability ($\alpha$), highest occupied molecular orbital energy ($\epsilon_{\text{HOMO}}$), and lowest unoccupied molecular orbital energy ($\epsilon_{\text{LUMO}}$). In contrast, for the GEOM dataset, the regression task focuses on predicting the energy. The QM9 dataset was obtained from the PyTorch Geometric (PyG) library and used in its original form. The GEOM dataset was sourced from the

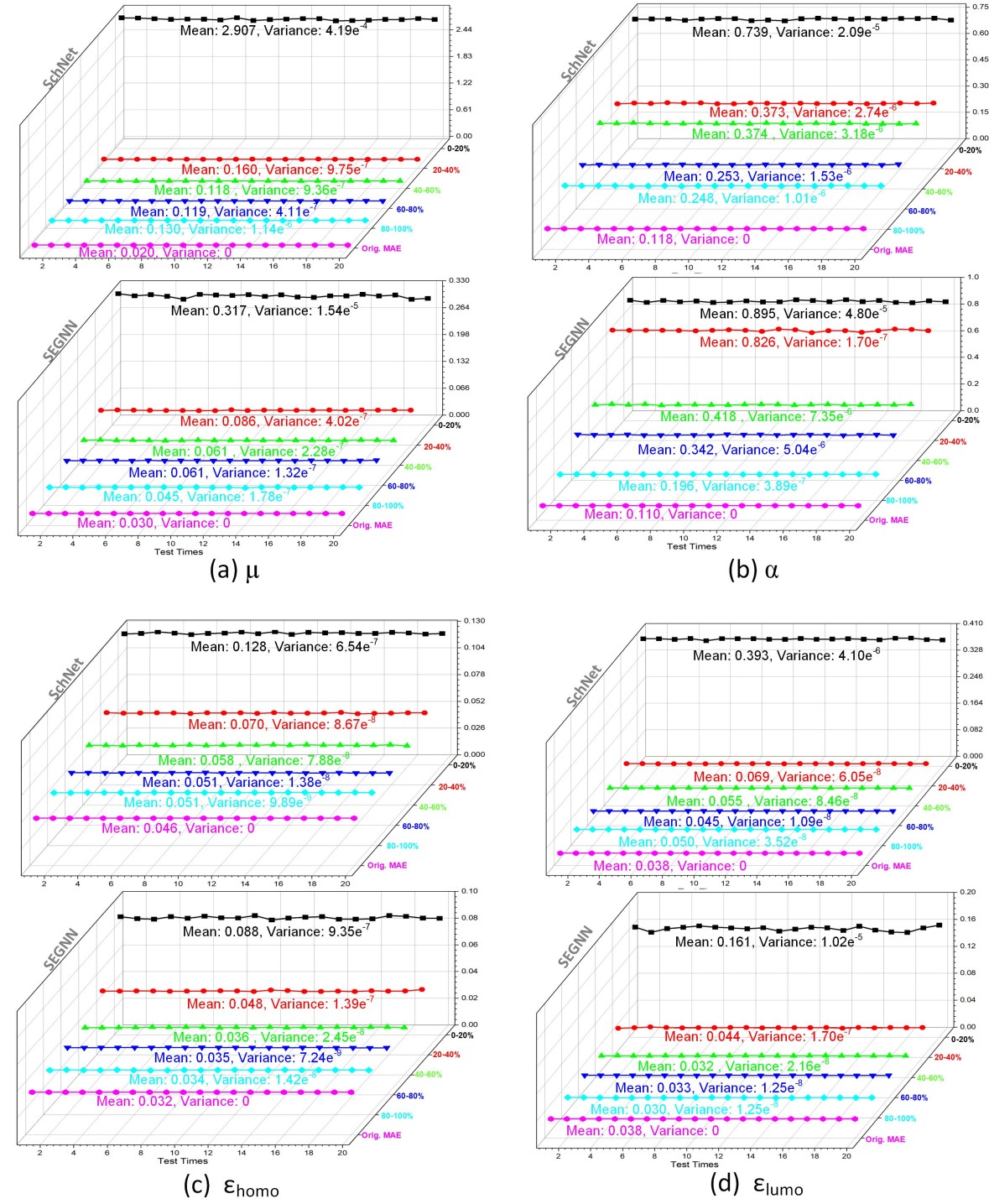

*Figure 9.* The visualization of MAE prediction on $\mu$, $\alpha$, $\epsilon_{\text{HOMO}}$, and $\epsilon_{\text{LUMO}}$ with randomly mask 10% edges within the top $0-20\%$, $20-40\%$, $40-60\%$, $60-80\%$, and $80-100\%$ distant range, respectively. The results on four properties jointly indicate the correlation between edge distance and corresponding importance, while edges within the same annulus have similar importance.

official GEOM paper (Axelrod & Gomez-Bombarelli, 2022), and we preprocessed it by removing all conformers except the lowest-energy one for each molecule.

As for the backbone models, on QM9 dataset, we parameterize SchNet and DimeNet models with the pre-trained weight

*Table 6.* Hyperparameter Search Space: To ensure fairness and reproducibility, we present the parameter search space used in our experiments. All models were evaluated under a range of different hyperparameter settings, and the best-performing results are reported.

| Explainer | Setting | Search Space |
|---|---|---|
| RISE | Loss Weights | $\lambda_{pred} = 1, \lambda_{size} = 0, \lambda_{ent} = 0$ |
| | Training Epochs | 50, 100, 300 |
| GNNExplainer | Loss Weights | $\lambda_{pred} = 1, \lambda_{pred} \in \{0.1, 0.5, 1.0\}, \lambda_{ent} \in \{0.5, 1\}$ |
| | Training Epochs | 50, 100, 300 |
| PGExplainer | Loss Weights | $\lambda_{pred} = 1, \lambda_{pred} \in \{0.1, 0.2, 0.5, \mathbb{Z} \in [1, 10]\}, \lambda_{ent} \in \{0.05, 0.1, 0.2\}$ |
| | Training Epochs | 50, 100, 300, 500 |
| LRI-Bernoulli | Loss Weights | $\lambda_{pred} = 1, \lambda_{pred} \in \{0.5, \mathbb{Z} \in [1, 10], 20\}, \lambda_{ent} \in \{0.01, 0.05, 0.1, 0.2\}$ |
| | Training Epochs | 50, 100, 300, 500 |

provided by the PyG (Fey & Lenssen, 2019) library with their original settings. The SEGNN model was trained on QM9 following the official settings described in the paper. For the GEOM dataset, model settings were maintained the same as the experiments on QM9. The cut-off for all models was set to $5.0$ on GEOM. The backbone models are first trained on $50,000$ molecules from GEOM dataset, which is then filtered and split into $10,000$ molecules for training explanation models, and $1,000$ for testing. All models are trained independently on a single NVIDIA RTX A6000 GPU.

We configure our RISE method as described in Sec. 4. For the baseline methods, we adjust the weights of the loss function, which given as

$$\mathcal{L} = \lambda_{pred}\mathcal{L}_{pred} + \lambda_{size}\|\mathcal{M}_{soft}\|_1 + \lambda_{ent}\mathbb{H}[M_{soft}], \tag{12}$$

where the $\mathcal{L}_{pred}$ is the prediction loss. The search space of weights and training epochs are shown in Table 6. After training, baseline methods select the top $k$ edges based on their mask values. The original LRI method lacks the size loss, which makes it learn a dense edge mask with all values close to $1$. Therefore, we incorporate the size loss into its original loss function as well.

## I. Additional Experimental Results

*Table 7.* Resulting MAE (the lower the better) using **DimeNet** as the backbone model on the QM9 dataset. The setup from the original work employs a cut-off distance of $5\mathring{A}$, leading to an average of 279 edges per graph. The average number of edges preserved is indicated in parentheses. The best and second best results are respectively highlighted in red and blue.

| Explainer | $\mu$ | | | $\alpha$ | | |
|---|---|---|---|---|---|---|
| | 0.4 | 0.5 | 0.6 | 0.4 | 0.5 | 0.6 |
| GNNExplainer | **0.897(29.9%)** | **0.795(39.9%)** | **0.667(50.0%)** | 25.873(29.9%) | 17.769(39.9%) | **10.073(50.0%)** |
| PGExplainer | 1.202(31.3%) | 1.410(48.4%) | 1.423(57.1%) | 56.00(30.0%) | 112.81(40.0%) | 208.19(50.0%) |
| LRI-Bernoulli | 7.932(33.3%) | 6.087(44.1%) | 5.265(52.8%) | **15.963(34.7%)** | **15.142(41.2%)** | 13.740(54.3%) |
| **RISE (ours)** | **0.697(25.2%)** | **0.570(37.8%)** | **0.458(49.8%)** | **5.004(25.2%)** | **1.485(37.8%)** | **1.087(49.8%)** |

| Explainer | $\epsilon_{HOMO}$ | | | $\epsilon_{LUMO}$ | | |
|---|---|---|---|---|---|---|
| | 0.5 | 0.6 | 0.7 | 0.5 | 0.6 | 0.7 |
| GNNExplainer | **1.505(39.8%)** | **0.995(50.0%)** | **0.493(69.8%)** | 1.055(39.8%) | **0.882(50.0%)** | **0.368(69.8%)** |
| PGExplainer | **3.712(42.6%)** | 3.596(49.7%) | 3.268(71.3%) | 1.039(40.0%) | 1.002(50.0%) | 0.950(70.0%) |
| LRI-Bernoulli | 3.646(40.0%) | 3.547(50.0%) | 3.128(70.0%) | **0.944(40.0%)** | 0.927(50.0%) | 0.911(70.0%) |
| **RISE (ours)** | 5.091(37.6%) | **2.727(49.1%)** | **1.486(65.9%)** | **0.749(37.8%)** | **0.731(47.5%)** | **0.197(66.4%)** |

*Table 8.* Resulting MAE (the lower the better) using **SchNet** and **DimeNet** as the backbone models on the GEOM dataset. The setup from the original work employs a cut-off distance of $5\mathring{A}$, leading to an average of 981 edges per graph. The average number of edges preserved is indicated in parentheses. The best and second best results are respectively highlighted in red and blue, or both in red if they are very close and the lower one has more edges preserved.

| Explainer | SchNet | | | DimeNet | | |
|---|---|---|---|---|---|---|
| | 0.4 | 0.5 | 0.6 | 0.4 | 0.5 | 0.6 |
| GNNExplainer | 0.105(30.0%) | 0.106(40.0%) | 0.102(50.0%) | **10.397(30.0%)** | **4.224(40.0%)** | **2.292(50.0%)** |
| PGExplainer | **0.071(25.8%)** | **0.084(39.5%)** | **0.097(53.2%)** | 47.605(20.0%) | 41.645(30.0%) | 29.949(50.0%) |
| LRI-Bernoulli | 17.535(20.0%) | 17.532(40.0%) | 17.531(50.0%) | 33.018(20.0%) | 24.633(30.0%) | 12.018(50.0%) |
| **RISE (ours)** | **0.098(19.6%)** | **0.096(31.2%)** | **0.082(43.2%)** | **0.845(17.3%)** | **0.198(27.3%)** | **0.081(40.5%)** |

## I.1. RISE Performance: DimeNet on QM9 Dataset

On the QM9 dataset, we additionally provide the results using DimeNet as the backbone model in Table 7. The results indicate that RISE outperforms all baselines in most cases across different budgets. For instance, at $B = 0.6$, RISE achieves the lowest MAE for $\mu$, $\alpha$, and $\epsilon_{\text{LUMO}}$ predictions, as well as the second-lowest MAE for $\epsilon_{\text{HOMO}}$, while preserving fewer edges. These results demonstrate the superiority of RISE in capturing meaningful interactions.

## I.2. RISE Performance: Comparison on GEOM Dataset

To demonstrate the generalizability of our method, we also conducted experiments on the GEOM dataset, which contains the molecules related to experimental data in the biophysics, physiology, and physical chemistry domains. The quantitative results are shown in Table 8. RISE consistently outperforms or is at least on par with existing explanation methods. These findings highlight the robustness of our approach in adapting to diverse molecular structures and learning meaningful edge attributions.

