# OpenReview forum: "RISE: Radius of Influence based Subgraph Extraction for 3D Molecular Graph Explanation"
_ICML.cc/2025/Conference — ICML 2025 poster_

### Official Review · Reviewer_FVPY · 2025-03-12

**Overall Recommendation:** 2

**Summary:**

The paper proposes a new instance-level method to “explain” the predictions of 3D molecular GNNs. Following earlier work, this is formulated as an optimization problem over subgraphs, where the objective is to minimize the loss of predictive power when removing edges, under a given budget of edges. The author's main observation is that the edges between close-by atoms generally should affect the performance of 3D GNNs more. They support this with experiments by deleting random edges of a certain distance and showing a stronger drop in performance the shorter the deleted edges. Based on this they formulate the optimization problem using a “radius of influence”, which is a radius around each atom beyond which the edges are being cut. This contrasts with previous literature, that optimizes soft masks without any distance-based inductive bias.
The authors highlight two major benefits: 1) The radius of influence is smooth and therefore optimizable with gradient-based methods, which is claimed to eliminate the need for thresholding soft masks into hard masks. 2) The budget can be enforced strictly without relying on penalty terms.
The authors test the method by measuring the drop in predictive power when a certain percent of edges are removed and show that their method performs best compared to previous methods. They also claim that their method is the only one that preserves known important features such as covalent bonds.


### update after rebuttal

Thank you for the clarifications. I think the experiments are well conducted, but I am still unconvinced as to what the usefulness exactly is. The authors say, “ The main purpose is not to identify novel chemical interactions unknown to chemists, but rather to understand how ML models make decisions. This is particularly important for scientific applications, where explainability is crucial for domain scientists to trust ML models.” Unfortunately, at the current point in time, I do not trust the ML models more based on the explanations given by RISE. Showing that the model is most sensitive to covalent bonds is expected by the design of the networks, which have a strong near-sightedness bias built in.

However, I can imagine that my opinion would change if the authors could present less trivial cases where the method can pick up known non-trivial interactions, which would make me believe that the model learns physical many-body interactions. Another convincing experiment would be if the method can explain a negative case, so a case where the ML model fails, and the explainer shows that the model pays too much attention to some unnecessary/unphysical edge. This way the absence of weird edges could make me trust the model more. It would also expand the usefulness of the method to let practitioners find failure points and correct them in a targeted fashion.

The idea of the annuli for long-range interactions makes sense but needs to be proven. Again, I think this has a lot of potential to be impactful, but at the current point I am not convinced enough and therefore keeping my score.

**Claims And Evidence:**

The claim that atoms close to each other have a stronger influence is generally accepted and even enforced by 3D GNN architectures using a cutoff function that decreases a message's magnitude with distance. Since RISE is building on this observation it is generally a sensible idea, which is supported by the QM9 and GEOM experiments.  However, the central claim that the subgraphs found by RISE are more interpretable is less well supported, or even defined what is meant by interpretable. The biggest piece of evidence is given in figure 4 and 8, where RISE only leaves the covalent bonds, whereas the other methods also include other, less interpretable edges. However, all approaches include the covalent bonds, so the fact that there are additional edges found by the other approaches could also just be due to the bigger budget afforded to them.

**Essential References Not Discussed:**

The paper does a good job of discussing the most relevant explainability literature. I think a section explaining the key pieces of the used 3D GNNs, in particular how messages are constructed would be a good addition.

**Experimental Designs Or Analyses:**

Discussed above

**Methods And Evaluation Criteria:**

The authors use the drop-off in QM9 and GEOM performance given a certain budget of edges as a figure of merit and show that their method retains better performance than previous methods. This shows that the subgraphs picked out by RISE are more predictive on average on the given datasets. However, it is not clear to me if this would still be the case for systems where long-range and non-local effects are important since RISE would be forced to generate very large radii of influence to find these interactions, thereby including many potentially unnecessary edges, whereas previous methods could find only a few important edges that mediate the effects.

**Other Comments Or Suggestions:**

In my eyes, the two most pressing suggestions is to contextualize the vision of the paper better. Why do we care about these explainers? Are there any unexpected explanations made by RISE that a chemist wouldn't have thought of easily? Show some example where RISE picks out a non trivial edge that correlates to interesting physics.
The paper mentions several times that the number of edges in a 3D GNN grows exponentially. I am not sure if I misunderstand what the authors are trying to say, however, the number of edges in a fully connected graph is quadratic in the number of nodes, not exponential.
Another small suggestion to improve the flow of the paper is that the paper talks about search spaces (line 48) before explaining that these explanation methods are formulized as optimization problems. Adding a short sentence in the beginning to give this context would be helpful

**Other Strengths And Weaknesses:**

The paper is original and the presentation is good

**Questions For Authors:**

How would RISE deal with long-range and non-local interactions?
Are there any non-trivial explanations found by RISE beyond covalent bonds that can be seen as a known physical interaction?
What is the overall vision with these explanation approaches? What are they useful for? What kind of actionable insights are we hoping to gain?

**Relation To Broader Scientific Literature:**

The paper is an adaptation of existing ideas to 3D GNNs

**Theoretical Claims:**

The authors say that other methods have to threshold soft edges which induces a mismatch between optimization and resulting graphs, whereas RISE does not require this thresholding. As far as I understand RISE also includes a form of soft masks due to equation 7, such that there is still a mismatch in the mask and the optimization.
The other claim about exactly enforceable budgets seems solid though.

---

> ### Author Rebuttal · Authors · 2025-03-31
>
> Thank you so much for your detailed and constructive comments! We provide our responses here.
>
> ## Claims And Evidence
>
> > Interpretability of explanation
> - The radii of influence can be intepreted as **the spatial extent within which an atom can significantly affect its surroundings**.
>
> > Additional edges by other approaches due to the bigger budget
> - There is a misunderstanding here. **The budget is exactly the same for all methods.** This is stated in the caption of Fig. 8. We will include clearer statements in the revision to further emphasize this.
> - In terms of interpretability, **the edges found by all explanation methods are directed as in their original works**. RISE successfully finds both directions of the covalent bonds, while other methods find one direction of the covalent bonds and some less interpretable edges. We will include this clarification in the revision.
>
> ## Methods And Evaluation Criteria
>
> > Long-range interactions
> - The current form of our method is mainly for small molecules as we note in our discussion for future works in Sec. 5.  **Our approach also has the potential to be extended to scenarios with long-range interactions.** We provide some key insights below and will include such discussions in Appendix in the next revision.
> - We could allow multiple radii of influence for nodes, represented as annular regions that collectively adhere to a predefined budget. For example, for a given atom, the model could identify edges with distances in the ranges [0.0, 0.3] and [0.6, 0.8] as important. This extension aligns well with chemical principles, as different types of interactions—such as covalent bonding, van der Waals forces, and electrostatic interactions—tend to dominate at specific distance ranges. **As a result, to capture important long-range interactions, only relevant annuli will be included; it won't result in unnecessary edges.**
>
> ## Theoretical Claims
>
> > Relaxation to Soft Masks
> - **There might be some misunderstanding here.** Indeed, both RISE and baseline methods relax the weights of edges from discrete values to continuous values. However, **there is a fundamental difference**. **The loss function of RISE is a function of radii; it can be extremely close to discrete values with the relaxation, e.g., Eq. (7). Similar approaches, e.g. applying Eq. (7) to edge/node masks, won't change the minima of the loss function of baseline methods, still resulting in mismatch.**
> - **RISE optimizes the radius masks, which are naturally continuous.** In Eq. (7), we reformulate the radius masks to be edge masks to allow gradient-based optimization. **As explained in sentences above Eq. (7), we can apply a function such that the edge masks will be extremely close to $0$ or $1$.**
> - Consider a node with three neighbors of distances $0.5$, $0.6$, and $0.7$, respectively. Suppose the optimized radius is $0.55$. **The resulting edge masks using Eq. (7) with $k=1,000$ are $1.00$, $1.93\times 10^{-22}$, and $7.18\times 10^{-66}$**, respectively .
> - Additionally, we present below the percentage of edge masks in different ranges resulting from real experiments on RISE and GNNExplainer, respectively. The closer the edge mask values are to $0$ or $1$, the smaller the mismatch between the mask and the binary optimization objective. **It can be seen clearly that the mismatch is almost negligible for RISE, but very severe for GNNExplainer.**
> |Edge Mask Value| GNNExplainer(%)|RISE(%)|
> |-|-|-|
> |[0, 0.1] and [0.9, 1.0]|0.003|99.314|
> |[0, 0.2] and [0.8, 1.0]|11.463|99.438|
> |[0, 0.3] and [0.7, 1.0]|54.508|99.741|
> Experiment constructed on the first 1000 molecules in QM9 test dataset for lumo using SchNet. On average, there are 314 edges per graph.
> ## Essential References Not Discussed
>
> > Discussion of 3D GNNs
> - **We have provided a discussion of different 3D geometric GNNs, including the ones used in this paper, in Appendix E.** We will include more details, such as how messages are constructed, in the next revision.
>
> ## Other Comments Or Suggestions:
>
> > The need of explanation
> - **In the introduction, we have briefly introduced the motivation and elaborated on the need of 3D GNN explanation methods.**
> -  We will include further discussion in the revision. **The main purpose is not to identify novel chemical interactions unknown to chemists, but rather to understand how ML models make decisions**. This is particularly important for scientific applications, where explainability is crucial for domain scientists to trust ML models.
>
> > The number of edges
> - We do not mean that the number of edges is $O(C^N)$ ($N$ is the number of nodes) but rather a term to describe the large number of edges 3D GNNs have. **We will change to the word "rapidily" in the next revision.**
>
> > Search spaces before introducing optimization
> - We will make necessary modifications.
>
> ## Questions For Authors:
> The questions have similar responses to the content of 'Other Comments Or Suggestions'. Please refer to the section above.

---

### Official Review · Reviewer_JV7x · 2025-03-13

**Overall Recommendation:** 4

**Summary:**

This contribution introduces RISE (Radius of Influence based Subgraph Extraction), an innovative explanatory approach for 3D geometric Graph Neural Networks (GNNs) in molecular learning.

RISE's principal contribution is the allocation of a "radius of influence" to each atom (node). This delineates the confined area where message passing encapsulates the most significant spatial and structural interactions for the model's predictions. RISE reconfigures 3D networks into directed proximity graphs (DPGs), whereby each node possesses a designated radius that governs the establishment of edges. By optimizing radii of impact, RISE extracts subgraphs that accurately reflect the model's predictions and are chemically interpretable.

The submission evaluates RISE across several 3D GNN architectures (SchNet, DimeNet, and SEGNN) and datasets (QM9 and GEOM), demonstrating its constant superiority over established explanation techniques, such as GNNExplainer, PGExplainer, and LRI-Bernoulli. Significantly, RISE generates chemically interpretable explanatory subgraphs that correspond with genuine chemical linkages, whereas current methodologies frequently give chemically uninterpretable outcomes.

## Update after rebuttal
The authors provided detailed discussions to address my concerns. I appreciate their effort and would like to retain my original score: accept this submission.

**Claims And Evidence:**

The claims made in the submission are well supported by empirical evidence.

Claim 1: Proximity (distance) determines the importance of message passing in 3D GNNs. To support this claim, the author conducted two experiments: The first one is removing edges from different distance-based annular bins and measuring the model performance. Secondly, randomly removing edges within each annulus to assess the variance in importance. The results in Table 1 clearly show that "removing closer annuli leads to a more significant drop in MAE, as values in most cells strictly decrease compared to the previous row." Figure 5 further demonstrates that "edges at similar distances have comparable importance, as indicated by the relatively flat trend lines with minimal fluctuation."

Claim 2: RISE outperforms existing explanation methods for 3D GNNs. This claim is supported by comprehensive experimental results across multiple backbone models (SchNet, DimeNet, and SEGNN) and datasets (QM9 and GEOM). Tables 2 and 3 show that "RISE consistently outperforms all baselines across various budgets" for most molecular properties and model configurations. The authors even note that they set "the baseline models' budgets... such that they strictly preserve more edges than RISE, making the comparison even more advantageous for the baselines," which strengthens the credibility of their results.

The third major claim is that RISE produces chemically interpretable explanatory subgraphs, unlike existing methods. This claim is supported by visualizations in Figure 4, which clearly show that "only RISE yields chemically interpretable results that conform to interpretable chemical structures." The authors provide a concrete example with the ethane molecule (CH₃CH₃), where "the radii of influence from our experiments assign the C of interest with a radius of 1.532 and the H of interest with a radius of 1.171," allowing RISE to extract the chemical bonds precisely.

**Essential References Not Discussed:**

N.A.

**Experimental Designs Or Analyses:**

The experimental methodology and analytical procedures presented in this submission are sufficient to demonstrate the effectiveness of RISE.

Two complementary approaches effectively examine the relationship between proximity and message passing significance in 3D. This experimental protocol appropriately isolates distance effects while maintaining control over confounding variables, thereby testing the proximity-importance hypothesis with methodological rigor.

The experimental outcomes, presented in Table 1 and Figure 5, demonstrate that "removing closer annuli leads to a more significant drop in MAE" (page 7) and that "edges at similar distances have comparable importance" (page 7). These findings provide substantial empirical support for the theoretical basis of RISE and its radius-of-influence conceptualization.

Methodological fairness is ensured by configuring "the baseline models' budgets... such that they strictly preserve more edges than RISE" (page 7), thereby introducing a conservative bias against the proposed method. This experimental constraint strengthens the validity of the comparative analysis by demonstrating RISE's superior performance despite operational disadvantages in edge preservation capacity.

The visualization analysis of explanatory subgraphs in Figure 4 offers qualitative evidence supporting RISE's capacity to generate chemically meaningful interpretations. The authors observe that "under a small budget, when explanation methods can preserve only a limited number of edges, RISE is the only method that selectively retains edges corresponding exclusively to chemical bonds" (page 8). This observation underscores the practical relevance of RISE for molecular science applications.

The experimental methodology and analytical procedures presented in this manuscript provide compelling evidence for RISE's effectiveness in explaining 3D molecular GNNs through rigorously designed and systematically executed evaluations.

**Methods And Evaluation Criteria:**

The methodological approaches and assessment criteria employed in this submission are well-suited and thoughtfully constructed.

The submission formulates the instance-level graph explanation, described as identifying "a subgraph $G^S \subseteq G$ that is important to the target $Y$." The authors effectively highlight the shortcomings of current techniques when implemented with 3D GNNs.

The RISE methodology demonstrates considerable theoretical rigor. By recasting 3D graphs as directed proximity graphs (DPGs), where "each node $v_i$ has an associated radius $0 \leq r_i$, and a directed edge $e_{i\rightarrow j}$ exists if and only if $d_{ij} < r_i$, where $d_{ij}$ is the distance between $v_i$ and $v_j$," the authors create a framework that corresponds well with established physical principles of molecular interactions. This approach acknowledges that "interactions between nodes separated by large distances are typically negligible due to the rapid decay of force magnitudes."

The evaluation strategy is appropriate and direct. The use of MAE as the primary performance indicator offers a straightforward measurement of how effectively the explanatory subgraph maintains the model's predictive accuracy.

Visual representations of the explanatory subgraphs offer compelling qualitative evidence of RISE's capacity to generate chemically meaningful interpretations. The researchers demonstrate that "under a small budget, when explanation methods can preserve only a limited number of edges, RISE is the only method that selectively retains edges corresponding exclusively to chemical bonds," representing a significant advancement for molecular science applications.

**Other Comments Or Suggestions:**

N.A.

**Other Strengths And Weaknesses:**

N.A.

**Questions For Authors:**

1. The paper focuses on relatively small molecular graphs. How would RISE scale to larger molecular systems that would exhibit long-range interaction?
2. How does RISE compare with attention-based explanation methods that might be adapted for 3D GNNs? Are there insights from attention mechanisms that could be incorporated into RISE?
3. Have you investigated the robustness of RISE explanations to small perturbations in molecular geometry?

**Relation To Broader Scientific Literature:**

This submission effectively positions its contributions within the domains of molecular graph learning, graph neural networks, and explainable artificial intelligence scholarship.

Building on established literature concerning the limitations of molecular representations, the authors articulate that "chemical behaviors and biological functions of molecules are largely determined by their 3D geometric structures" (page 1) while simultaneously recognizing contemporary advancements in three-dimensional graph learning methodologies.

The work extends the foundations established by previous explanation frameworks, such as GNNExplainer and PGExplainer. The authors identify a substantial methodological gap in the current literature, noting that "existing approaches struggle to effectively explain 3D GNNs" (page 2). They appropriately acknowledge the previously published research specifically addressing 3D GNN explanation while critically assessing its methodological constraints and limitations.

**Theoretical Claims:**

The theoretical foundations presented in this submission regarding RISE demonstrate logical consistency.

The introduction of directed proximity graphs (DPGs) as a mathematical extension of 3D graphs presents a rigorous framework. This formulation establishes a suitable mathematical structure for representing molecular graphs with variable influence radii.

The authors' assertion that "any 3D graph constructed based on node radii can be viewed as a directed proximity graph" and that "a 3D graph constructed based on a cut-off distance (the same radius for all nodes) is a proximity graph under uniform radii" (page 5) follows directly from the DPG definition and exhibits proper mathematical derivation.

The paper establishes that RISE maintains "consistency in optimization, as it does not require converting discrete values into continuous ones for optimization and does not include penalty or regularization terms to promote discreteness or enforce the budget" (page 5,6). This claim receives adequate support through the formulation of RISE, which optimizes continuous influence radii directly rather than relaxing binary masks to continuous values.

The bound provided to illustrate inconsistencies in existing methods, "$L(Y; \Phi(G^S)) \leq L(Y; \Phi(X, M_{soft} \odot A)) + L(\Phi(X, M_{soft} \odot A); \Phi(X, M \odot A))$" (page 4), correctly identifies the optimization objective discrepancy that remains unaddressed during the optimization phase in soft mask-dependent methods.

---

> ### Author Rebuttal · Authors · 2025-03-31
>
> Thank you so much for your efforts reviewing our work. We respond to your questions below.
> ## Questions For Authors:
> > Extension to larger systems with long-range interactions
> - Even in larger atomic systems, short-range interactions typically dominate chemical bonding and molecular stability. Covalent bonds, hydrogen bonds, and van der Waals forces are strongest at short distances, meaning **our method remains highly applicable and effective in most cases**.
> - However, there are scenarios where long-range interactions play a crucial role. For example, in proteins, tertiary and quaternary structures depend heavily on long-range interactions such as salt bridges, hydrophobic packing, and π-π stacking. **We have acknowledged this in the discussion of future work in Sec. 5.**
> - **Our approach also has the potential to be extended to such scenarios.** Instead of using a single value as the radius mask, we could allow multiple radii of influence, represented as annular regions that collectively adhere to a predefined budget. For example, for a given atom, the model could identify edges with distances in the ranges [0.0, 0.3] and [0.6, 0.8] as important. This extension aligns well with chemical principles, as different types of interactions—such as covalent bonding, van der Waals forces, and electrostatic interactions—tend to dominate at specific distance ranges.
>
> > Attention-based explanation methods
> - There are some attention-based methods in the literature, with Graph Attention Networks (GAT) being a representative. **Attention-based methods are ante-hoc**, e.g. [1]. Ante-hoc methods are built into the model itself, making it inherently interpretable (like based on evolving attention scores) while **our work's focus is on post-hoc explanation of any existing 3D GNNs** that use radius cut-off graphs. The scope of our work and that of attention-based explanation are different, so there is not a direct comparison.
> - In terms of incorporating insights from attention mechanisms, attention-based GNNs (e.g. GAT) provide explicit edge importance scores for each node with respect to all other nodes. This is different from subgraph extraction, where we try to find the important edges (substructures) with respect to the entire graph. Therefore, we think that attention mechanism might be incorporated into RISE as a joint optimization process. We jointly optimize attention weights and explanation masks, where the explanation seeks to refine or filter the attention. **Although this is a very interesting perspective, it is out of the scope of our current work.**
>
> > Small perturbations in molecular geometry
> - This is a good point! We did not test RISE under small perturbations in molecular geometry in the original version, because **Small perturbations will alter the ground truth values of chemical properties, making evaluation intractable.** Since the exact chemical property values after perturbation are unknown (intractable for us to compute each time a small perturbation is introduced), it is not possible to faithfully assess or compare different explanation methods in such cases.
> - Assuming minor perturbations in atomic positions do not alter chemical properties, we conducted a robustness test on the same 4 molecules as in the qualitative results in Appendix D (discussed in the last paragraph of Sec. 4.2). Results indicate that when adding normal Gaussian noise to a small fraction (1/5) of atoms, RISE maintains explanation stability. While perturbations affect radii of influence, the binary edge mask remains unchanged, which means subgraphs extracted by RISE remain the same. **This demonstrates the robustness of RISE under small perturbations** (and for sure, again, if we assume minor perturbations in atomic positions do not alter chemical properties.)
>
> [1] Interpretable and Generalizable Graph Learning via Stochastic Attention Mechanism, Siqi Miao et al., ICML 2022

---

> > ### Comment · Reviewer_JV7x · 2025-04-08
> >
> > Thanks for your detailed responses, which address all my concerns. I have no further comments.

---

> > > ### Author Response · Authors · 2025-04-08
> > >
> > > Thank you for your response and your kind, positive feedback on our work. We're delighted to hear that you're satisfied with our reply.

---

### Official Review · Reviewer_4kQJ · 2025-03-13

**Overall Recommendation:** 4

**Summary:**

This paper proposes a novel explanation method that localizes interpretability within each node’s immediate 3D neighborhood. By defining a "radius of influence," the approach constrains message passing to spatially and structurally relevant subgraphs. This enhances interpretability while maintaining alignment with the physical dependencies in molecular applications.

## update after rebuttal
The author's response has resolved many of my questions, and I will increase the score.

**Claims And Evidence:**

This paper provides a detailed analysis of the differences between 2D and 3D models, proposes a graph explanation method for the 3D domain, and demonstrates its superiority through experiments.

**Essential References Not Discussed:**

No

**Experimental Designs Or Analyses:**

The experiments compare the proposed RISE with some 3D baselines. However, from the paper, it can be seen that the proposed 3D model is an extension of the 2D model. Shouldn't an experiment be designed to compare the differences between the 2D and 3D models and examine the advantages of the 3D model in terms of interpretability?

**Methods And Evaluation Criteria:**

Yes

**Other Comments Or Suggestions:**

Section 4.1 is missing the bolded Results.

**Other Strengths And Weaknesses:**

None

**Questions For Authors:**

None

**Relation To Broader Scientific Literature:**

Not found

**Theoretical Claims:**

I am so sorry that I am not an expert in this field. Based on the reasoning from 2D to 3D, these formulas might be correct, but I am not entirely sure.

---

> ### Author Rebuttal · Authors · 2025-03-31
>
> Thank you very much for your valuable feedback! We provid our responses below.
> ## Experimental Designs Or Analyses
> > Comparison with baselines on 2D GNNs
> - **First of all, we want to emphasize that our work is not just an extension of its 2D counterparts.** Our work reformulates 3D graphs as directed proximity graphs (DPG), and based on this formulation, we determine the radii of influence. This is a principled approach specifically designed for 3D GNNs based on their unique characteristics.
> - We are not sure whether you are suggesting that we compare our method with 2D explanation baselines, even though they are designed for 2D GNNs, or that we evaluate our method on 2D GNNs.
> - **If you are suggesting a comparison with baselines for 2D GNN explanation, we have already included them in our work.** GNNExplainer and PGExplainer are the two franchise works for 2D GNN explanation. We have also compared our method with LRI, the only available explanation method designed for 3D GNNs.
> - **If you are suggesting an evaluation on 2D GNNs, our method is not applicable to them, as it explicitly considers the properties of 3D geometric graphs and 3D GNNs.** 2D topological GNNs cannot be formulated as directed proximity graphs, and there is no concept of distance or radius in 2D topological graphs. Our method aims to find the radii of influence for 3D geometric GNNs.
> ## Other Comments Or Suggestions
> > Missing bolded 'Results'
> - Thanks for pointing this out! We will revise it in the next version.

---

### Official Review · Reviewer_U9Dy · 2025-03-15

**Overall Recommendation:** 2

**Summary:**

The paper presents RISE, a method for explaining 3D molecular GNNs by identifying key substructures using a radius of influence for each atom. RISE formulates the explanation process as an optimization problem that finds a compact, chemically interpretable subgraph while maintaining prediction fidelity. Instead of soft edge masks, RISE optimizes the radius of influence for each atom, ensuring consistency between the explanation and the model’s learning process.
The authors evaluate RISE on QM9 and GEOM datasets, comparing it against GNNExplainer, PGExplainer, and LRI-Bernoulli. The results show that RISE produces more interpretable explanations, capturing actual chemical substructures like functional groups and bonds while maintaining high explanation fidelity.

**Claims And Evidence:**

1. Claim: **Unlike traditional 2D molecular graphs, 3D molecular graphs introduce implicit dense edges based on spatial proximity, making existing explainability methods ineffective.** However, the experiments in this paper show that RISE doesn't significantly outperform GNNExplainer and PGExplainer. Since both baselines are very old, the motivation of this paper -- existing explainability methods ineffective on 3D graph, doesn't seem to be valid.

**Essential References Not Discussed:**

[1] Integrating Explainability into Graph Neural Network Models for the Prediction of X-ray Absorption Spectra, 2023

This paper also evaluated on QM9. How do you compare your method with it?

**Experimental Designs Or Analyses:**

1. The paper demonstrates that distance is a key factor in message-passing but does not explore alternative explanation strategies or compare different distance-based formulations.
2. Experiment is not thorough. See "Claims And Evidence".

**Methods And Evaluation Criteria:**

1. While the idea of using spatial relationships in 3D GNN explanations is valuable, the approach is mainly a reformulation of standard explainability techniques rather than a fundamentally new method.
2. RISE is compared against GNNExplainer, PGExplainer, and LRI-Bernoulli. The comparisons are reasonable, but the lack of additional subgraph-level explainability baselines limits the scope of the evaluation since this is a subgraph-based method. (For example, SubgraphX, MAGE)
3. Unlike other methods that produce random subgraphs, RISE extracts explanations that correspond to actual chemical bonds, making them useful for domain experts. But authors are encouraged to provide results showing that by comparing with the existing method.

**Other Comments Or Suggestions:**

N/A

**Other Strengths And Weaknesses:**

N/A

**Questions For Authors:**

N/A

**Relation To Broader Scientific Literature:**

Recognizing Learning Differences in 3D GNNs: Rather than treating graphs as fixed node-edge structures, RISE treats them as node-radius systems, which better aligns with molecular interactions.

**Theoretical Claims:**

No

---

> ### Author Rebuttal · Authors · 2025-03-31
>
> Thanks for reviewing our work. We respond below.
>
> ## Claims And Evidence
> - The baselines are given budgets **preserving more edges than RISE, favoring them in comparison**.
> - Despite this, RISE **consistently** shows strong performance.
>
> > Baselines old; existing methods ineffective on 3D graph
> - **The issues of applying existing methods to 3D GNNs have also been noted in Sec. 2 of the LRI paper [1].**
> - **There are not too many explanation methods that can be used.**
>     - The provided reference **MAGE is not comparable to instance-level methods (our work)**, as it is **model-level**. MAGE does not generate subgraphs but identifies key motifs **across the whole dataset** for a prediction class.
>     -  The other reference, SubgraphX, is older than some baselines like PGExplainer and LRI. **Nevertheless, we have included its results (see "Methods and Evaluation Criteria")**.
>
>
>
> ## Methods And Evaluation Criteria
> > Not a fundamentally new method
> - Our method is **not just a reformulation** of existing methods.
> - **Eq. (1) defines a well-established general framework for subgraph extraction.** Many works [1-4] reformulate Eq. (1) to Eq. (4) for gradient-based optimization with budget and sparsity constraints.
> - We reformulate Eq. (1) to Eq. (5) using **directed proximity graphs (DPG) tailored for 3D graphs**. **This is the first and only reformulation to 3D GNNs that is directly optimizable using radii of influence. This is fundamentally different from Eq. (4), which requires additional constraints to address issues from soft mask relaxation.**
>
> > Comparison with subgraph-level explainability baselines
> - **The provided reference may not be directly comparable to our method.**
>     - **MAGE is a model-level explanation method** that identifies the most influential motifs **across the entire dataset** for a given class label. Our method is instance-level, identifying **a subgraph for each graph**.
>     - SubgraphX searches through various node combinations, and the search space for hard mask optimization scales exponentially with the number of nodes. **Our method and all baselines relax hard masks to enable efficient gradient-based optimization.** Consequently, **SubgraphX is 5× slower than our method on QM9, with an even larger gap for larger molecules**.
>
> - Nevertheless, SubgraphX is still instance-level; we compare it with RISE. **The results below clearly show that RISE outperforms SubgraphX** (metric is MAE, lower, better). We present results on the first 1,000 molecules from the QM9 test data due to time constraints. We will include the full results in the revision if you feel necessary.
>     - **Beyond time complexity, SubgraphX shares the same limitation as all other node-masking methods (see Appendix C).** Its exponential scaling prevents extension to edge masking.
>
> |Method|μ(0.3)|μ(0.4)|μ(0.5)|α(0.3)|α(0.4)|α(0.5)|
> |-|-|-|-|-|-|-|
> |SubgraphX|10.844|10.528|10.354|2.550|2.632|2.816|
> |RISE|0.482|0.320|0.134|1.983|1.311|0.525|
>
> |Method|HOMO(0.3)|HOMO(0.4)|HOMO(0.5)|LUMO(0.3)|LUMO(0.4)|LUMO(0.5)|
> |-|-|-|-|-|-|-|
> |SubgraphX|0.387|0.282|0.272|0.610|0.557|0.540|
> |RISE|0.375|0.190|0.095|0.474|0.251|0.114|
>
> > RISE extracts meaningful results
> - **We have already provided several qualitative results in Appendix D**, as discussed in the last paragraph of Sec. 4.2. If you believe it is necessary, we will include more in the next revision.
>
>
> ## Experimental Designs Or Analyses
> > Different distance-based formulations
> - Our work is motivated by the finding that distance is a key factor in message passing for 3D geometric graphs. **No existing explanation method uses a distance-based formulation**.
>
> - That said, we are unsure if you suggest designing alternative distance-based formulations for comparison with RISE. **Our work is among the first to explore 3D GNN explanation. RISE is currently the best method we can propose based on our insights. Our findings can inspire the community to develop more geometry-oriented subgraph extraction methods.**
>
>
>
> ## Essential References Not Discussed
> > Comparison with the provided reference
> - **The scope of the provided reference [5] is very different from our work.**
>     - **[5] does not explain 3D geometric graphs but rather 2D graphs.**
>         - "*Molecular graphs were generated from the SMILES strings of the molecules*" (p2, 1st paragraph under Methods, 2nd last sentence)
>     - While they also use QM9, their focus is on explaining models predicting the X-ray absorption spectrum (XAS). In contrast, we propose a general method for 3D GNN explanation and evaluate it across multiple quantum-level properties.
>
>
> [1]Interpretable Geometric Deep Learning via Learnable Randomness Injection
>
> [2]Parameterized explainer for graph neural network
>
> [3]GNES: Learning to explain graph neural networks
>
> [4]Stratified GNN Explanations through Sufficient Expansion
>
> [5]Integrating Explainability into Graph Neural Network Models for the Prediction of X-ray Absorption Spectra

---

### Decision · Program_Chairs · 2025-05-01

**Decision:**

Accept (poster)

**Comment:**

This paper introduces RISE, a method to explain the decision of 3D GNN by locating the subgraphs which their decision is based on. The reviewers are essentially split regarding this paper, with two reviewers supporting acceptance and two rejection. Overall, while the reviewers agree the paper is well executed, some of the reviewers are skeptical regarding (a) the importance of the problem in general and (b) the need for methods targeting 3D GNN specifically, in comparison with existing "2D GNN" explainability methods. The comparisons in the paper show some improvement but perhaps not substantial enough

I tend to support acceptance. Re (a) I know that chemist *are* interested in explainable models, and re (b) comparisons do show a consistent advantage in terms of accuracy. The proposed method seems to be the first explainability method proposed for 3D GNNs, so potentially this could be a nice contribution to the community.